# REINFORCEMENT LOGIC RULE LEARNING FOR TEMPORAL POINT PROCESSES

## ABSTRACT

We aim to learn a set of temporal logic rules to explain the occurrence of temporal events. Leveraging the temporal point process modeling and learning framework, the rule content and rule weights are jointly learned by maximizing the likelihood of the observed noisy event sequences. The proposed algorithm alternates between a master problem, where the rule weights are updated, and a subproblem, where a new rule is searched and included. The formulated master problem is convex and relatively easy to solve, whereas the subproblem requires searching the huge combinatorial rule predicate and relationship space. To tackle this challenge, we propose a neural search policy to learn to generate the new rule content as a sequence of actions. The policy parameters will be trained end-to-end using the reinforcement learning framework, where the reward signals can be efficiently queried by evaluating the subproblem objective. The trained policy can be used to generate new rules, and moreover, the well-trained policies can be directly transferred to other tasks to speed up the rule searching procedure in the new task. We evaluate our methods on both synthetic and real-world datasets, obtaining promising results.

## 1 INTRODUCTION

Understanding the generating process of events with irregular timestamps has long been an interesting problem. Temporal point process (TPP) is an elegant probabilistic model for modeling these irregular events in continuous time. Instead of discretizing the time horizons and converting the event data into time-series event counts, TPP models directly model the inter-event times as random variables and can be used to predict the *time-to-event* as well as the future *event types*. Recent advances in neural-based temporal point process models have exhibited superior ability in event prediction (Du et al., 2016; Mei & Eisner, 2017). However, the lack of interpretability of these black-box models hinders their applications in high-stakes systems like healthcare.

In healthcare, it is desirable to summarize medical knowledge or clinical experiences about the disease phenotypes and therapies to a collection of logic rules. The discovered rules can contribute to the sharing of clinical experiences and aid to the improvement of the treatment strategy. They can also provide explanations to the occurrence of events. For example, the following clinical report

*"A 50 years old patient, with a chronic lung disease since 5 years ago, took the booster vaccine shot on March 1st. The patient got exposed to the COVID-19 virus around May 12th, and afterward within a week began to have a mild cough and nasal congestion. The patient received treatment as soon as the symptoms appeared. After intravenous infusions at a healthcare facility for around 3 consecutive days, the patient recovered... "*

contains many clinical events with timestamps recorded. It sounds appealing to distill compact and human-readable temporal logic rules from these noisy event data. In this paper, we propose an efficient *reinforcement temporal logic rule learning* algorithm to automatically learn these rules from event sequences. See Fig. 1 for a better illustration of the types of temporal logic rules we aim to discover, where the logic rules are in disjunctive normal form (i.e., OR-of-ANDs) with temporal ordering constraints.

Our proposed reinforcement rule learning algorithm builds upon the *temporal logic point process* (TLPP) models (Li et al., 2020), where the intensity functions (i.e., occurrence rate) of events are informed by temporal logic rules. TLPP is intrinsically a probabilistic model that treats the temporal

logic rules as *soft constraints*. The learned model can tolerate the uncertainty and noisiness in events and can be directly used for future event prediction and explaination. Given this TLPP modeling framework, our reinforcement rule learning algorithm jointly learns the rule content (i.e., model structures) and rule weights (i.e., model parameters) by *maximizing the likelihood* of the observed events. The designed learning algorithm alternates between solving a convex *master problem*, where the continuous rule weight parameters are easily optimized, and solving a more challenging *subproblem*, where a new candidate rule that *has the potential to most improve the current likelihood* is discovered via reinforcement learning. New rules are progressively discovered and included until by adding new rules the objective will not be improved.

Specifically, we formulate the rule discovery *subproblem* as a *reinforcement learning* problem, where a *neural policy* is learned to efficiently navigate through the *combinatorial* search space to search for a good explanatory temporal logic rule to add. The constructed neural policy emits a distribution over the pre-specified logic predicate and temporal relation libraries, and generates the logic variables as actions in a sequential way to form the rule content. The generated rules can be of various lengths. Once a temporal logic rule is generated, a terminal *reward* signal can be efficiently queried by evaluating the current *subproblem objective*

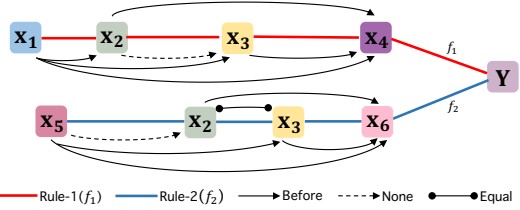

Figure 1: Example of temporal logic rules: $f_1 : Y \leftarrow x_1 \wedge x_2 \wedge x_3 \wedge x_4 \wedge (x_1 \ Before \ x_2) \wedge (x_2 \ None \ x_3) \wedge (x_3 \ Before \ x_4)$, $f_2 : Y \leftarrow x_5 \wedge x_2 \wedge x_3 \wedge x_6 \wedge (x_5 \ None \ x_2) \wedge (x_2 \ Equal \ x_3) \wedge (x_3 \ Before \ x_6)$. $None$ means no temporal order constraints.

using the generated rule, which is computationally expedient, without the need to worry about the insufficient reward samples. The neural policy is gradually improved by a risk-seeking policy gradient to learn to generate rules to optimize the subproblem objective, which is rigorously formulated from the dual variables of the master problem so as to search for a rule that has the potential to best improve the current likelihood.

This proposed reinforcement logic rule learning algorithm has the following advantages: 1) We utilize differentiable policy gradient to solve the temporal logic rule search subproblem. All the policy parameters can be learned end-to-end via policy gradient using the subproblem objective as reward. 2) Domain knowledge or grammar constraints for the temporal logic rules can be easily incorporated by applying specific dynamic masks to the rule generative process at each time step. 3) The memories of how to search through the rule space have been encoded in the policy parameters. The well-trained neural policies for each subproblem can be directly transferred to similar rule learning tasks to speed up the computation in new tasks, where we don't need to learn rules from scratch.

**Contributions** Our main contributions have the following aspects:

*i)* We propose an efficient and differentiable reinforcement temporal logic rule learning algorithm, which can automatically discover temporal logic rules to predict and explain events. Our method will add flexibility and explainability to the temporal point process models and broaden their applications in scenarios where interpretability is important.

*ii)* All the well-trained neural policies in solving each subproblem can be readily transferred to new tasks. This fits the continual learning concept well. The quality of the rule search policies can be continually improved across various tasks. For a new task, we can utilize the preceding tasks' memories even though we cannot get access to the old training data. We empirically evaluated the transferability of our neural policies and achieved promising results.

*iii)* Our discovered temporal logic rules are human-readable. Their scientific accuracy can be easily judged by human experts. The discovered rules may also trigger experts in thinking. In our paper, we considered a real healthcare dataset and mined temporal logic rules from these clinical event data. We invited doctors to verify these rules and incorporated their feedback and modification into our experiments.

## 2 RELATED WORK

**Temporal point process (TPP) models.** TPP models can be characterized by the intensity function. The modeling framework boils down to the design of various intensity functions to add the model flexibility and interpretability (Mohler et al., 2011). Recent development in deep learning has

significantly enhanced the flexibility of TPP models. (Du et al., 2016) proposed a neural point process model, named RMTPP, where the intensity function is modeled by a Recurrent Neural Network. (Mei & Eisner, 2017) improved RMTPP by constructing a continuous-time RNN. (Zuo et al., 2020) and (Zhang et al., 2020a) recently leveraged the self-attention mechanism to capture the long-term dependencies of events. Although flexible, these neural TPP models are black-box and are hard to interpret. To add transparency, (Zhang et al., 2020b) used Granger causality as a latent graph to explain point processes, and the structures are jointly learned via gradient descent. However, Granger causality is still limited to the mutual triggering patterns of events. Recently, (Li et al., 2020) proposed an explainable Temporal Logic Point Process (TLPP), where the intensity function is built on the basis of temporal logic rules. TLPP model enables one to perform symbolic reasoning for events however their rules are required to be pre-specified. A follow-up work (Li et al., 2022) designed a column generation type of temporal rule learning algorithm. However, their subproblems are solved by enumeration, which will be intractable for long temporal logic rules and their search memories cannot be reused for future tasks. By contrast, our proposed algorithm makes the subproblem differentiable and the trained neural policies can be reused and transferred.

**Logic rule learning methods.** Learning logic rules without temporal relation constraints has been studied from various perspectives. Recently, (Wang et al., 2017) tried to learn an explanatory binary classifier using the Bayesian framework. SATNet (Wang et al., 2019a) transformed rule mining into a SDP-relaxed MaxSAT problem. Attention-based methods (Yang & Song, 2019) were also introduced. Neural-LP (Yang et al., 2017) provided the first fully differentiable rule mining method based on TensorLog (Cohen, 2016), and (Wang et al., 2019b) extended Neural-LP to learn rules with numerical values via dynamic programming and cumulative sum operations. In addition, DRUM (Sadeghian et al., 2019) connected learning rule confidence scores with low-rank tensor approximation. (Dash et al., 2018; Wei et al., 2019) introduced a column generation (i.e., branch and price) type of MIP algorithm to learn the logic rules. However, all these above-mentioned logic learning methods cannot be directly applied to event sequences with timestamps. By contrast, we designed a differentiable algorithm to learn temporal logic rules from event sequences.

**Learning model structures via reinforcement learning (RL).** RL provided a promising approach to automatically finding the best-fitting model structures, which inspired us to apply it to rule learning. For example, in AutoML, the famous NAS (Zoph & Le, 2017) trained a recurrent neural network by RL to design the architectures of deep neural networks and has achieved comparable performance with the human-designed models. Similar ideas have been adopted to aid the design of the explainable machine learning models. For example, the RL algorithm has been successfully introduced to learn the causal graph to explain the data (Zhu et al., 2020). Recently, the RL algorithm has been used in symbolic regression (Petersen, 2021; Landajuela, 2021), which aims to learn the set of compact mathematical expressions to explain the dynamics of complex dynamic systems. In our paper, we customized the RL algorithm to learn the temporal logic rules to explain the event sequences.

## 3 BACKGROUND

### 3.1 TEMPORAL POINT PROCESSES

Given an event sequence $\mathcal{H}_t = \{t_1, t_2, \ldots, t_n | t_n < t\}$ up to $t$, which yields a counting process $\{N(t), t \geq 0\}$, the dynamics of the TPP can be characterized by conditional intensity function, denoted as $\lambda(t|\mathcal{H}_t)$. By definition, we have $\lambda(t|\mathcal{H}_t)dt = \mathbb{E}[N([t, t + dt])|\mathcal{H}_t]$, where $N([t, t + dt])$ denotes the number of points falling in an interval $[t, t + dt]$. By some simple proof (Rasmussen, 2018), one can express the joint likelihood of the events $\mathcal{H}_t$ as

$$p(\{t_1, t_2, \ldots, t_n | t_n < t\}) = \prod_{t_i \in \mathcal{H}_t} \lambda(t_i|\mathcal{H}_{t_i}) \cdot \exp\left(-\int_0^t \lambda(\tau|\mathcal{H}_\tau)d\tau\right). \tag{1}$$

The TPP modeling boils down to the design of intensity functions and the model parameters can be learned by maximizing the likelihood. Recent neural-based event models start modeling $\lambda_\theta(t \mid \mathcal{H}_t)$ as a deep learning model, such as RNN, which greatly increases the model flexibility but the learned model is hard to interpret. In this paper, we consider TLPP, where the intensity is constructed by temporal logic rules.

## 3.2 Temporal Logic Rules

**Predicate** Define a set of entities $\mathcal{C} = \{c_1, c_2, ..., c_n\}$. The *predicate* is defined as the *property* or *relation* of entities, which is a logic function that is defined over the set of entities, i.e., $x(\cdot) : \mathcal{C} \times \mathcal{C} \cdots \times \mathcal{C} \mapsto \{0, 1\}$. For example, $Smokes(c)$ is the property predicate and $Friend(c, c')$ is the relation predicate.

**Temporal Logic Rule** A *first-order logic rule* is a logical connectives of predicates, such as $f : \forall c, \forall c', Smokes(c') \leftarrow Friend(c, c') \wedge Smokes(c)$. A temporal dimension is added to the predicates. The temporal logic rule is a logic rule with temporal ordering constraints. For example, $f : \forall c, Covid(c, t_3) \leftarrow SymptomsAppear(c, t_2) \wedge ExposedToVirus(c, t_1) \wedge Before(t_1, t_2)$. For discrete events, we consider three types of temporal relations, $Before(t_1, t_2) = \mathbb{1}\{t_1 - t_2 < 0\}, After(t_1, t_2) = \mathbb{1}\{t_1 - t_2 > 0\}, Equal(t_1, t_2) = \mathbb{1}\{t_1 = t_2\}$. We also treat the temporal relation as the temporal predicate, which is a boolean variable. Formally, we consider the following general temporal logic rule (for simplicity, we omit the entity index $c$ and assume all the predicates are defined for the same entity),

$$f : Y(t_y) \leftarrow \underbrace{\bigwedge_{u \in \mathcal{X}_f} X_u(t_u)}_{\text{property predicates}} \underbrace{\bigwedge_{u, u' \in \mathcal{X}_f} \mathcal{T}_{uu'}(t_u, t_{u'})}_{\text{temporal relations}} \tag{2}$$

where $Y(t_y)$ is the head predicate evaluated at time $t_y$, $\mathcal{X}_f$ is the set of predicates defined in rule $f$, $\mathcal{T}_{uu'}$ denotes the temporal relation of predicate $u$ and $u'$ that can take any mutually exclusive relation from the set $\{Before, After, Equal, None\}$. Note that $None$ indicates there is no temporal relation constraint between predicate $u$ and $u'$. $t_y$, $t_u$, and $t'_u$ are the occurrence times associated with the predicates.

## 3.3 Temporal Logic Point Process

The *grounded* temporal predicate $x(c, t)$, such as $Smokes(c, t)$, generates a sequence of discrete events $\{t_1, t_2, \dots\}$, with the time that the predicate becomes 1 (i.e., True) is recorded.

**Logic-informed intensity function** The main idea of TLPP (Li et al., 2020) is to construct the intensity functions using temporal logic rules. TLPP considers complicated logical dependency patterns, which enables symbolic reasoning for temporal event sequences. Gven a rule as Eq. (2), TLPP builds the intensity function conditional on history. Only the effective combinations of the historical events that makes the body condition True will be collected to reason about the intensity of the head predicate. We introduce a logic function $g_f(\cdot)$ to check the body conditions of $f$. $g_f(\cdot)$ can also incorporate temporal decaying kernels to capture the decaying effect of the evidence like Hawkes process. The logic-informed feature $\phi_f$ gathers the information from history, which is computed as

$$\phi_f(t) = \sum_{u \in \mathcal{X}_f} \sum_{\{t_u\} \in \mathcal{H}_t^u} g_f\left(\{t_u\}_{u \in \mathcal{X}_f}\right) \tag{3}$$

where $\mathcal{H}_t^u$ indicates the historical event specific to predicate $u$ up to $t$. Suppose there is a rule set $\mathcal{F}$ that can be used to reason about $Y$. For each $f \in \mathcal{F}$, one can compute the features $\phi_f(t)$ as above. Assume that the rules are connected in disjunctive normal form (OR-of-ANDs). TLPP models the intensity of the head predicate $\{Y(t)\}_{t \geq 0}$ as a log-linear function of the features,

$$\lambda(t \mid \mathcal{H}_t) = \exp\left(b_0 + \sum_{f \in \mathcal{F}} w_f \cdot \phi_f(t)\right) \tag{4}$$

where $\boldsymbol{w} = [w_f]_{f \in \mathcal{F}} \geq 0$ are the learnable weight parameters associated with each rule, and $b_0$ is the learnable base intensity. All the model parameters can be learned by maximizing the likelihood as defined in Eq. (1). Note that given this intensity model, the likelihood is convex with respect to $\boldsymbol{w}$ (Fahrmeir et al., 1994).

## 4 Reinforcement Temporal Logic Rule Learning

Our goal is to jointly learn the set of OR-of-ANDs temporal logic rules and their weights by MLE. Each rule has a general form (2). To discover each rule, the algorithm needs to navigate through the combinatorial space considering all the combinations of the property predicates and their temporal relations. Moreover, each rule can have various lengths and the computational complexity grows exponentially with the rule length. To tackle this challenging problem, we propose a RL search

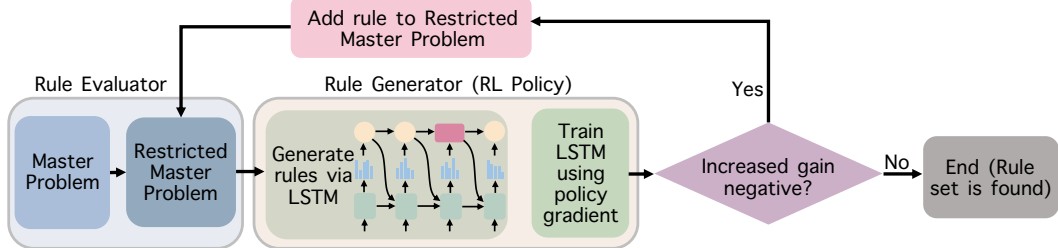

Figure 2: Overall Learning Framework: alternating process between rule generator and rule evaluator

algorithm to discover new rules one-by-one. The overall learning framework is shown in Fig. 2where the algorithm alternates between a master problem (rule evaluator) and a subproblem (rule generator).

### 4.1 OVERALL LEARNING OBJECTIVE: A REGULARIZED MLE

We formulate the overall model learning problem as a regularized MLE problem, where the objective function is the log-likelihood with a rule set complexity penalty, i.e.,

$$\text{Original Problem}: \quad \boldsymbol{w}^*, b_0^* = \arg\min_{\boldsymbol{w}, b_0} -\ell(\boldsymbol{w}, b_0) + \Omega(\boldsymbol{w}) \quad s.t. \quad w_f \geq 0, \quad f \in \bar{\mathcal{F}} \quad (5)$$

where $\bar{\mathcal{F}}$ is the complete rule set, and $\Omega(\boldsymbol{w})$ is a convex regularization function that has a high value for "complex" rule sets. For example, we can formulate $\Omega(\boldsymbol{w}) = \lambda_0 \sum_{f \in \bar{\mathcal{F}}} c_f w_f$ where $c_f$ is the rule length.

#### 4.1.1 RESTRICTED MASTER PROBLEM: CONVEX OPTIMIZATION

The above original problem is hard to solve, due to that the set of variables is exponentially large and can not be optimized simultaneously in a tractable way. We therefore start with a restricted master problem (RMP), where the search space is much smaller. For example, we can start with an empty rule set, denoted as $\mathcal{F}_0 \subset \bar{\mathcal{F}}$. Then we gradually expand this subset to improve the results, this will produce a nested sequence of subsets $\mathcal{F}_0 \subset \mathcal{F}_1 \subset \cdots \subset \mathcal{F}_k \subset \cdots$. For each $\mathcal{F}_k$, $k = 0, 1, \ldots$, the restricted master problem is formulated by replacing the complete rule set $\bar{\mathcal{F}}$ with $\mathcal{F}_k$:

$$\text{Restricted Master Problem}: \quad \boldsymbol{w}^*_{(k)}, b^*_{0,(k)} = \arg\min_{\boldsymbol{w}, b_0} -\ell(\boldsymbol{w}, b_0) + \Omega(\boldsymbol{w}) \quad s.t. \ w_f \geq 0, \ f \in \mathcal{F}_k. \quad (6)$$

Solving the RMP corresponds to the evaluation of the current candidate rules. All rules in the current set will be reweighed. The optimality of the current solution can be verified under the principle of the *complementary slackness* for convex problems, which in fact leads to the objective function of our subproblem. More proof can be found in the Appendix F.

#### 4.1.2 SUBPROBLEM: COMBINATORIAL PROBLEM

A subproblem is formulated to propose a new temporal logic rule, which can potentially improve the optimal value of the RMP most. Given the current solution $\boldsymbol{w}^*_{(k)}, b^*_{0,(k)}$ for the restricted master problem (6), a subproblem is formulated to minimize the increased gain:

$$\text{Subproblem:} \quad \min_{\phi_f} \ -\frac{\partial \ell(\boldsymbol{w}, b_0)}{\partial w_f} + \frac{\partial \Omega(\boldsymbol{w})}{\partial w_f} \Bigg|_{\boldsymbol{w}^*_{(k)}, b^*_{0,(k)}} \quad (7)$$

where $\phi_f$ is a rule-informed feature. We aim to search for a new rule so that the corresponding feature minimizes the above objective function. If the optimal subproblem value is *negative*, we will include the new rule to the set. Otherwise, if the optimal subproblem value is non-negative, we reach the optimality and we can stop the overall algorithm. However, this search procedure is also computationally expensive. It requires search rule structures to have a feature evaluation. In the following, we will discuss how to solve the subproblem more efficiently using reinforcement learning.

We want to emphasize here that, although the idea above as how to formulate the master and subproblems have been widely used in machine learning, including the gradient grafting algorithm for learning high-dimensional linear models (Perkins et al., 2003), column-generation algorithm for solving mixed integer programming (Savelsbergh, 2002; Nemhauser, 2012; Lübbecke & Desrosiers, 2005) and large linear programming (Demiriz et al., 2002), and for learning ordinal logic rules (Dash et al., 2018; Wei et al., 2019), our overall learning framework is gradient-based. For the master problem, we solve a convex optimization with continuous variables by gradient descent (or SGD). For

the subproblem, although it requires searching over the combinatorial space, we convert the problem into learning a neural policy where the policy parameters can be learned by policy gradient.

## 4.2 SOLVING SUBPROBLEMS VIA REINFORCEMENT LEARNING

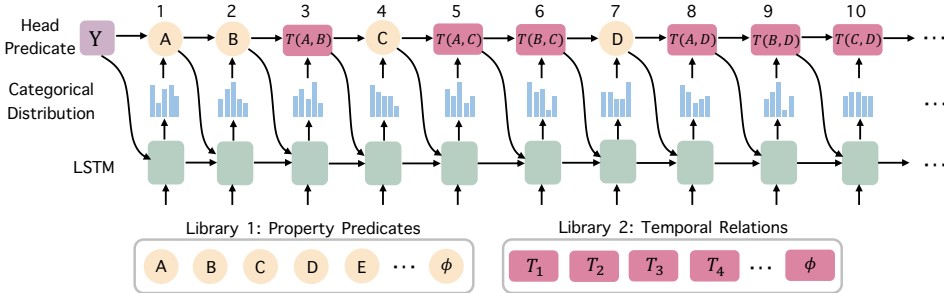

Figure 3: Illustration of generating a temporal logic rule using a neural policy (LSTM).

The subproblem formulated as Eq. (7) proposes a criterion to propose a new temporal logic rule. The subproblem itself is a minimization, which aims to attain the most negative increased gain. However, explicitly solving the subproblem requires enumerating all possible conjunctions of the input property predicates and all possible pairwise temporal relations among the selected predicates, which is extremely computationally expensive. In this paper, instead of trying to enumerate all the conjunctions, we propose to learn a neural policy by reinforcement learning to generate the best-explanatory rules.

### 4.2.1 GENERATING RULES WITH RECURRENT NEURAL NETWORK

We leverage the fact that the temporal logic rules can be represented as a sequence of "tokens" subject to some unique structures. Using the pre-ordered traversal trick, we parameterize the policy by an RNN or LSTM, combined with dynamic masks, to guarantee that the generated tokens can yield a valid temporal logic rule.

We generate the rules one token at a time according the pre-order traversal, as demonstrated in Fig. 3. Specifically, denote $s$ as the state, which is the embedding of the previously generated tokens. We model the policy as $\pi_\theta(a|s)$ with learnable parameter $\theta$, which quantifies the token selection probability given $s$. Each token/action can be chosen from the two predefined libraries: *i)* the property predicate libraries, and *ii)* the temporal relation libraries.

Given the head predicate, we generate the body predicates and their temporal relations in a sequential way. Every time a property predicate is generated, we need to consider its temporal relation with all the previously generated property predicates. Note that the temporal relation token can be None, which means there is no temporal relation constraints. All these generative prior knowledge can be incorporated as constraints by designing dynamic masks.

### 4.2.2 RISK-SEEKING POLICY GRADIENT

The standard policy gradient objective $J(\theta)$ is defined as an expectation. This is the desired objective for control problems in which one seeks to optimize the average performance of a policy. However, rule learning problems described in our paper are to search for best-fitting rules. For such problems, $J(\theta)$ may not appropriate, as there is a mismatch between the objective being optimized and the final performance metric.

We consider risk-seeking policy gradient like (Petersen, 2021; Landajuela, 2021), which proposed an alternative objective that aims to *maximize the best-case performance*. According to the original work (Landajuela, 2021; Petersen, 2021), we first define $R_\epsilon(\theta)$ as the $(1 - \epsilon)$-quantile of the distribution of rewards under the current policy. Then the new objective $J_{risk}(\theta; \epsilon)$ is given by:

$$J_{risk}(\theta; \epsilon) = \mathbb{E}_{\tau \sim \pi(\tau|\theta)} \left[ R(\tau) \mid R(\tau) \geq R_\epsilon(\theta) \right] \tag{8}$$

Then the risk-seeking policy gradient can be estimated using the roll-out samples, i.e.,

$$\nabla_\theta J_{risk}(\theta; \epsilon) \approx \frac{1}{\epsilon N} \sum_{i=1}^{N} \sum_{k=1}^{K} \left[ R^{(i)}(\tau) - \tilde{R}_\epsilon^{(i)}(\theta) \right] \cdot \mathbb{1}_{R^{(i)}(\tau) \geq \tilde{R}_\epsilon^{(i)}(\theta)} \nabla_\theta \log \pi_\theta(a_k^{(i)}|s_k^{(i)}) \tag{9}$$

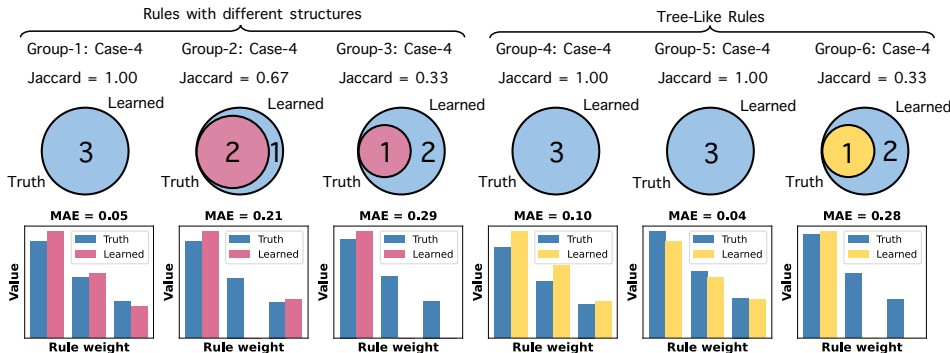

Figure 4: Rule discovery ability and rule weight learning accuracy of our proposed model based on Case-4 for all 6 groups. Blue one indicates ground truth rule and red/yellow one indicates learned rule in different groups.

where $N$ is the number of episodes, and $K$ is the length of tokens (actions). $\tilde{R}_\epsilon^{(i)}(\theta)$ is the empirical $(1 - \epsilon)$-quantile of the batch of rewards, and $\mathbb{1}$ returns 1 if condition is true and 0 otherwise. We use this estimated policy gradient to update the policy $\theta$.

$$\theta \leftarrow \theta + \alpha \nabla_\theta J_{risk}(\theta; \epsilon) \tag{10}$$

where $\alpha$ is the learning rate. Further, according to the maximum entropy reinforcement learning framework (Haarnoja et al.), a bonus can be added to the loss function proportional to the entropy to help the policy do the exploration.

## 5 EXPERIMENTS

### 5.1 SYNTHETIC DATA

We prepared 6 groups of synthetic event data, each group with a different set of ground truth rules. We considered the following baselines: TELLER (Li et al., 2022), policy gradient without risk seeking, and brute-force method (enumerating all possible rules).

**Accuracy and Scalability**   For each group, we further considered 5 cases, with the to-be-searched property predicate library being sized 8, 12, 16, 20, and 24, respectively. Note that only a small amount of the predicates will be in the true rules. Many of the predicates are redundant information and they will act as background predicates. We aim to test: 1) whether our reinforcement temporal logic learning algorithm can truly uncover the rules from the noisy variable set, 2) how accurate can the rule weights be estimated, 3) and how the performance will evolve if we gradually increase the variable set with more and more redundant variables.

The ground truth rules of different groups are with different length and various content structures. For some groups, each rule shares many common predicates in content, while for some groups, the designed rules are quite distinct in content. For example, the ground truth rules in group-$\{1, 2, 3\}$ are quite different in their content, and the ground truth rules in group-$\{4, 5, 6\}$ share many common predicates. On the other hand, in group-$\{1, 4\}$ and group-$\{2, 5\}$, the number of property predicates in one ground truth rule is 6 and 7 respectively. We set the number of property predicates of ground truth rules to be 8 in group-$\{3, 6\}$ to craft relatively long rules, especially considering intricate temporal relation at the same time. When uncovering ground truth rules in every group, we fix 2 predicates as prior knowledge and ask the algorithm to complete the temporal logic rules. Complete results for all the datasets can be found in Appendix I.

In Fig. 4, we reported the learning results for the case with predicate library size 20 for all groups (with different rule content). We used 1000 samples of event sequences as training data. Each plot in the top row uses a Venn diagram to show the true rule set and the learned rule set, from which the Jaccard similarity score (area of the intersection divided by the area of their union) is calculated. Our proposed model discovered almost all the ground truth rules. Each plot in the bottom compares the true rule weights with the learned rule weights, with the Mean Absolute Error (MAE) reported. Almost all the truth rule weights are accurately learned and the MAEs are quite low. In group-3 and group-6, we crafted long and complex rules with 8 body property predicates and various temporal

relations, which yield an extremely huge search space, but our model still discovered almost all the ground truth rules.

Fig. 5 illustrates the Jaccard similarity score and MAE for *all cases* in *all 6 groups* using 1000 samples. For all 6 groups, as the number of predicates in the predicate set increases, the Jaccard similarity scores decrease slightly and the MAE increases slightly, but it is still within an acceptable range. This is because as the number of redundant predicates increases, the search space expands exponentially and the complexity of searching is dramatically increased. But if the number of predicates in the predicate set is appropriate and the samples are sufficient, our model is very stable and reliable.

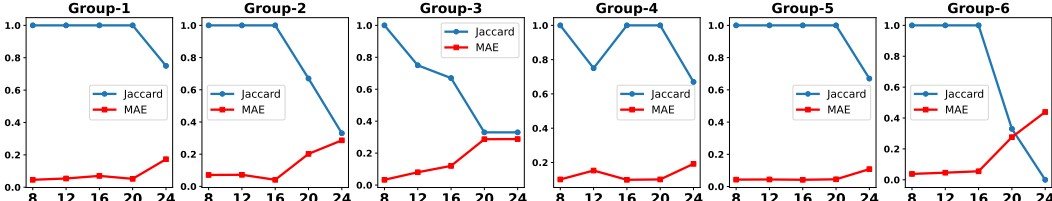

Figure 5: Jaccard similarity score and MAE for all 6 groups. X-axis indicates the predicate library size and Y-axis indicates the value of Jaccard similarity and MAE

**Computational Efficiency**    As shown in Fig.6 (left), we displayed the curve of the log-likelihood function versus the running time for our method and baselines. Our method uses the risk-seeking policy gradient in solving subproblems (refer to the flat-line period in figure). As a comparison, vanilla (normal) policy gradient still optimize the expectation of the reward to solve the subproblem. For TELLER, it uses enumerative search to generate new rules in subproblems although adopts the depth-first type of heuristic to try to append predicates to important short rules to generate long rules. As for the brute-force method, it first considers all the one-body-predicate rules to optimize the likelihood by learning the rule importance, then adds all the two-body-predicate rules to the model, and so on. From the results, we see that our method uncovers the ground truth rules faster and more accurately compared with all the baselines in the long-run. It is almost intractable for the brute-force method. Compared with the normal policy gradient method, the performance and accuracy of our model are also better, mainly because by using risk-seeking policy gradient, the model can focus learning only on maximizing best-case performance. We did more experiments to compare the normal policy gradient and risk-seeking policy gradient. Please refer to Appendix K for more details.

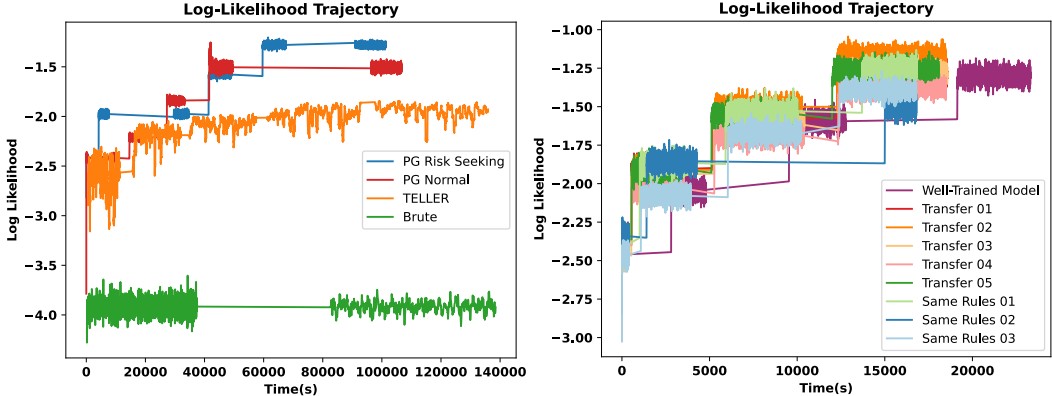

Figure 6: Log Likelihood Trajectory. Left: Compare our model (risk seeking policy gradient) with several baselines (normal policy gradient, TELLER, Brute force way). Right: Transfer experiments. First completely train a collection of rule generators and save the model parameters, then use the well-trained model to uncover new rules on different datasets. *Transfer ##* indicates the new datasets with ground truth rules that are slightly different compared with the datasets used by the well-trained model. *Same Rule ##* indicates the new datasets generated by same ground truth rules.

**Transferability**    Our well-trained neural search policies are also transferable. As shown in Fig. 6 (right), once we get a collection of well-trained policies in solving each subproblem, we can use it to train on other different datasets which are generated by the same or similar ground truth rules. The

results showed that this will speed up the process of solving subproblems and improve the efficiency and accuracy of uncovering ground truth rules in these datasets.

## 5.2 Healthcare Dataset

MIMIC-III is an electronic health record dataset of patients admitted to the intensive care unit (ICU) (Johnson et al., 2016). We considered patients diagnosed with sepsis (Saria, 2018; Raghu et al., 2017; Peng et al., 2018), since sepsis is one of the major causes of mortality in ICU. Previous studies suggest that the optimal treatment strategy is still unclear. It is unknown what is the optimal treatment strategy in terms of using intravenous fluids and vasopressors to support the circulatory system. There also exists clinical controversy about when and how to use these two groups of drugs to reduce the side effect for the patients. For this real problem, we implemented our proposed reinforcement temporal logic rule learning algorithm to learn explanatory rules and their weights and gain insight into this problem.

**Discovered temporal logic rules**  In Appendix C, we reported all the uncovered temporal logic rules and their weights learned by our algorithm. We use `LowUrine` and `NormalUrine` as the head predicates respectively. We also invited human experts (doctors) to check the correctness of these discovered rules and the doctors think most of these rules have clinical meaning and are consistent with the pathogenesis of sepsis. Doctors' modifications and suggestions for these algorithm-discovered rules are also provided. Experts think that Rule 1-9 capture the major lab measures, like low systolic blood pressure, high blood urea nitrogen and low central venous pressure (appeared in Rule 2), that usually emerge before extremely low urine. Rule 10-18 shed light on drug and treatment selection. For example, reflected in Rule 12, Crystalloid and Dobutamine together yields a weight of 0.4535 for patient's normal urine.

**Compared with baselines in event prediction**  We considered the following SOTA baselines: 1) Recurrent Marked Temporal Point Processes (RMTPP) (Du et al., 2016), the first neural point process (NPP) model, where the intensity function is modeled by a Recurrent Neural Network (RNN); 2) Neural Hawkes Process (NHP) (Mei & Eisner, 2017), an improved variant of RMTPP by constructing a continuous-time LSTM; 3) THP, Transformer Hawkes Process (Zuo et al., 2020), which leverages the self-attention mechanism to capture long-term dependencies and meanwhile enjoys computational efficiency; 3) Tree-Regularized GRU (TR-GRU) (Wu et al., 2018), a deep time-series model with a designed tree regularizer to add model interpretability; 5) Hexp (Lewis & Mohler, 2011), Hawkes Process with an exponential kernel; 6) Transformer (Vaswani et al., 2017), an advanced model which follows an encoder-decoder structure, but does not rely on recurrence and convolutions in order to generate an output. 7) TELLER (Li et al., 2022), alternating between master problem and subproblem (enumerative search) to learn logic rules; 8) PG-normal, neural search policy learned by normal policy gradient, without risk seeking.

We used mean absolute error (MAE) as the evaluation metrics for the event prediction tasks of the two head predicates: 1) `LowUrine` and 2) `NormalUrine`, which are evaluated by predicting the time when these events will occur. Lower MAE (unit is hour in this example) indicates better performance of model. The performance of our model and all baselines are compared in Tab. 1, from which one can observe that our model outperforms all the baselines in this experiment.

Table 1: Event prediction results.

| Method | LowUrine (MAE) | NormalUrine (MAE) |
|---|---|---|
| RMTPP | 1.989 | 2.170 |
| NHP | 1.784 | 1.977 |
| THP | 1.658 | 1.759 |
| TR-GRU | 1.627 | 1.692 |
| Hexp | 2.578 | 2.483 |
| Transformer | 1.704 | 1.872 |
| TELLER | 1.887 | 1.532 |
| PG-normal | 2.050 | 1.647 |
| **OURMETHOD** | **1.622** | **1.305** |

## 6 Conclusion

In this paper, we proposed a reinforcement temporal logic rule learning algorithm to jointly learn temporal logic rules and their weights from noisy event data. The proposed learning algorithm alternates between a rule generator stage and a rule evaluator stage, where a neural search policy is learned by risk-seeking gradient descent to discover new rules in rule generator stage. The use of the neural policy makes the subproblem differentiable, and the well-trained policies can be easily transferred to other tasks. We empirically evaluated our method on both synthetic and healthcare datasets, obtaining promising results.

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

## A  APPENDIX

In the following, we will provide supplementary materials to better illustrate our methods and experiments.

- B presents different ground truth rule structures and some experiment results on synthetic data.

- C presents all learned rules on MIMIC-III data and expert identification and revision suggestion.

- Section D provides the definitions of all types of temporal relations considered in our model.

- Section E and F elaborate on the necessary proofs, which justify our model and learning framework.

- Section G provides pseudocode to illustrate our rule generator policy, subproblem formulation, and the complete algorithm.

- Section H provides our computing infrastructure.

- Section I, J and K show the comprehensive experiments on synthetic data. In K we compared the performance between standard policy gradient and risk seeking policy gradient.

- Section L introduces the background on the MIMIC-III data and provides a list of the chosen predicates used in our real experiment. Section M presents more experiment results in terms of the modified rules by experts.

- Section N introduces another application using our proposed method, which is an experiment about understanding shoppers' purchase patterns given their eye fixation event data.

## B  ALL LEARNED RULES ON SYNTHETIC DATA

We did experiments on 6 groups of event data, each group with a different set of ground truth rules. For each group, we further considered 5 cases. To illustrate different rule structures in different structure, we listed all the discovered rules for Case-2 of all groups in Tab. 2, Tab. 3, and Tab. 4. Due to limited space, we did not show all learned rule content for all cases of all groups. Please refer to I for the complete results of experiments on synthetic data.

Table 2: Results of Case-2 for Group-1, Group-4. *Ground truth rules which are learned. * Ground truth rules which are not learned. * Rules that are wrongly learned.

| Group-1 Case-2 | Group-4 Case-2 |
|---|---|
| * Y ← A ∧ B ∧ C ∧ D ∧ E ∧ F
∧ A Before B ∧ B Before C
∧ C Before D ∧ D Before E
∧ E Before F | * Y ← A ∧ B ∧ C ∧ D ∧ E ∧ F
∧ A Before B ∧ B Before C
∧ C Before D ∧ D Before E
∧ E Before F |
| * Y ← C ∧ B ∧ F ∧ D ∧ E ∧ G
∧ C Before B ∧ B Before F
∧ F Before D ∧ D Before E
∧ E Before G | * Y ← A ∧ B ∧ C ∧ D ∧ E ∧ G
∧ A Before B ∧ B Before C
∧ C Before D ∧ D Before E
∧ E Before G |
| * Y ← C ∧ B ∧ G ∧ D ∧ E ∧ H
∧ C Before B ∧ B Before G
∧ G Before D ∧ D Before E
∧ E Before H | * Y ← A ∧ B ∧ C ∧ D ∧ E ∧ H
∧ A Before B ∧ B Before C
∧ C Before D ∧ D Before E
∧ E Before H |
| —- | * Y ← D ∧ B ∧ C ∧ A ∧ E ∧ H
∧ D Before B ∧ B Before C
∧ C Before A ∧ A Equal E
∧ E Equal H |

Table 3: Results of Case-2 for Group-2 and Group-5. *Ground truth rules which are learned. *Ground truth rules which are not learned. *Rules that are wrongly learned.

| Group-2 Case-2 | Group-5 Case-2 |
|---|---|
| *Y ← A ∧ B ∧ C ∧ D ∧ E ∧ F ∧ G
∧ A Before B ∧ B Before C
∧ C Before D ∧ D Before E
∧ E Before F ∧ F Equal G | *Y ← A ∧ B ∧ C ∧ D ∧ E ∧ F ∧ G
∧ A Before B ∧ B Before C
∧ C Before D ∧ D Before E
∧ E Before F ∧ F Equal G |
| *Y ← C ∧ B ∧ F ∧ D ∧ E ∧ G ∧ H
∧ C Before B ∧ B Before F
∧ F Before D ∧ D Before E
∧ E Before G ∧ G Equal H | *Y ← A ∧ B ∧ C ∧ D ∧ E ∧ F ∧ H
∧ A Before B ∧ B Before C
∧ C Before D ∧ D Before E
∧ E Before F ∧ F Equal H |
| *Y ← C ∧ B ∧ A ∧ D ∧ E ∧ H ∧ G
∧ C Before B ∧ B Before A
∧ A Before D ∧ D Before E
∧ E Before H ∧ H Equal G | *Y ← A ∧ B ∧ C ∧ D ∧ E ∧ F ∧ H
∧ A Before B ∧ B Before C
∧ C Before D ∧ D Before E
∧ E Before F ∧ F Before H |

Table 4: Results of Case-2 for Group-2 and Group-5. *Ground truth rules which are learned. *Ground truth rules which are not learned. *Rules that are wrongly learned.

| Group-3 Case-2 | Group-6 Case-2 |
|---|---|
| *Y ← A ∧ B ∧ C ∧ D ∧ E ∧ F ∧ G ∧ H
∧ A Before B ∧ B Before C
∧ C Before D ∧ D Before E
∧ E Before F ∧ F Equal G
∧ G Equal H | *Y ← A ∧ B ∧ C ∧ D ∧ E ∧ F ∧ G ∧ H
∧ A Before B ∧ B Before C
∧ C Before D ∧ D Before E
∧ E Before F ∧ F Equal G
∧ G Equal H |
| *Y ← C ∧ B ∧ F ∧ D ∧ E ∧ G ∧ H ∧ A
∧ C Before B ∧ B Before F
∧ F Before D ∧ D Before E
∧ E Before G ∧ G Equal H
∧ H Equal A | *Y ← A ∧ B ∧ C ∧ D ∧ E ∧ F ∧ G ∧ H
∧ A Before B ∧ B Before C
∧ C Before D ∧ D Before E
∧ E Before F ∧ F Equal G
∧ G Before H |
| *Y ← C ∧ B ∧ A ∧ D ∧ E ∧ H ∧ G ∧ F
∧ C Before B ∧ B Before A
∧ A Before D ∧ D Before E
∧ E Before H ∧ H Equal G
∧ G Equal F | *Y ← A ∧ B ∧ C ∧ D ∧ E ∧ F ∧ H ∧ G
∧ A Before B ∧ B Before C
∧ C Before D ∧ D Before E
∧ E Before F ∧ F Equal H
∧ H Before G |
| *Y ← A ∧ C ∧ H ∧ G ∧ B ∧ D ∧ E ∧ F
∧ A Before C ∧ C Before H
∧ H Before G ∧ G Before B
∧ B Before D ∧ D Before E
∧ E Before F | —- |

## C    ALL LEARNED RULES ON MIMIC-III DATA

The complete sets of learned rule on MIMIC-III data are shown in Tab. 5 and Tab. 6. We invite human experts to check our learned rules and provide several revision suggestion. Rules shown in these tables have been identified by experts as clinically meaningful and consistent with sepsis pathology. Some of the rules are directly learned by our method without any modification (marked with a blue asterisk). And some are basically consistent with clinical facts, but some contents of the rules are not pathological. These rules have been slightly modified by experts (marked with a red asterisk).

Table 5: Part of learned rules with `LowUrine` as the head predicate. *Rules that are clinically meaningful confirmed by experts. *Rules that are modified by experts.

| Weight | Rule |
|---|---|
| 0.0015 | *__Rule 1:__ LowUrine ← `LowSysBP` ∧ `HighBUN` ∧ `LowArterialpH`
∧ (`LowSysBP Equal HighBUN`)
∧ (`HighBUN Equal ArterialpH`) |
| 0.1846 | *__Rule 2:__ LowUrine ← `LowSysBP` ∧ `HighBUN` ∧ `LowCVP`
∧ (`LowSysBP Before HighBUN`)
∧ (`HighBUN Equal LowCVP`) |
| 0.1175 | *__Rule 3:__ LowUrine ← `LowSodium` ∧ `LowChloride` ∧ `HighCreatinine`
∧ (`LowSodium Equal LowChloride`)
∧ (`LowChloride Equal HighCreatinine`) |
| 0.8226 | *__Rule 4:__ LowUrine ← `LowSodium` ∧ `LowChloride` ∧ `LowSVR`
∧ (`LowSodium Equal LowChloride`)
∧ (`LowChloride Equal LowSVR`) |
| 0.3694 | *__Rule 5:__ LowUrine ← `LowSodium` ∧ `LowChloride` ∧ `HighBUN`
∧ (`LowSodium Before LowChloride`)
∧ (`LowChloride Before HighBUN`) |
| 0.0727 | *__Rule 6:__ LowUrine ← `HighArterialLactate` ∧ `HighSVR` ∧ `LowBUN` ∧ `LowArterialpH`
∧ (`HighArterialLactate Equal HighSVR`)
∧ (`HighSVR Equal LowBUN`)
∧ (`LowBUN Equal LowArterialpH`) |
| 0.5554 | *__Rule 7:__ LowUrine ← `HighArterialLactate` ∧ `HighSVR` ∧ `HighPotassium` ∧ `LowCVP`
∧ (`HighArterialLactate Equal HighSVR`)
∧ (`HighSVR Equal HighPotassium`)
∧ (`HighPotassium Equal LowCVP`) |
| 0.0546 | *__Rule 8:__ LowUrine ← `HighArterialLatectatepH` ∧ `HighSVR`
∧ `HighHCO3` ∧ `LowSodium`
∧ (`HighArterialLatectatepH Equal HighSVR`)
∧ (`HighSVR Equal HighHCO3`) ∧ (`HighHCO3 Equal LowSodium`) |
| 0.4935 | *__Rule 9:__ LowUrine ← `LowSysBP` ∧ `HighArterialLactate`
∧ `HighPotassium` ∧ `HighChloride` ∧ `HighBUN`
∧ (`LowSysBP Equal HighArterialLactate`)
∧ (`HighArterialLactate Equal HighPotassium`)
∧ (`HighPotassium Equal HighChloride`)
∧ (`HighChloride Equal HighBUN`) |

## D    DETAILED EXPLANATION OF TEMPORAL RELATION

In this paper, the temporal relation was defined among events. For any pairwise events, denoted as $A$ and $B$, there exist only three types of temporal relations, which can be grounded by their occurrence times, denoted as $t_A$ and $t_B$. See below Table 7 for illustrations.

The temporal relation of any two events will be treated as temporal ordering constraints and can be included in a temporal logic rule as Eq. (4). Note that when included in the rule, the temporal relation can be none, which indicates that there is no temporal relation constraint between the two events in order to satisfy the rule.

## E    PROOF OF THE LIKELIHOOD FUNCTION OF TLPP

The likelihood function of the TLPP is a straightforward result from TPP. Readers can refer to the proofs in (Rasmussen, 2018). To be self-contained, we will provide a sketch of proof here.

For a specific entity $c$, given all the events associated with the head predicate $(t_1^c, t_2^c, \dots) \in [0, t)$, the likelihood function is the joint density function of these events. Using the chain rule, the joint likelihood can be factorized into the conditional densities of each points given all points before it.

Table 6: All learned rules with `NormalUrine` as the head predicate. *Rules that are clinically meaningful confirmed by experts. *Rules that are modified by experts.

| Weight | Rule |
|---|---|
| 0.2231 | **\*Rule 10:**NormalUrine ← LowUrine ∧ Crystalloid ∧ Phenylephrine ∧ (LowUrine Equal Crystalloid) ∧ (Crystalloid Equal Phenylephrine) |
| 0.0773 | **\*Rule 11:**NormalUrine ← LowUrine ∧ Phenylephrine ∧ Dopamine ∧ (LowUrine Before Phenylephrine) ∧ (Phenylephrine Before Dopamine) |
| 0.4535 | **\*Rule 12:**NormalUrine ← LowUrine ∧ Crystalloid ∧ Dobutamine ∧ (LowUrine Equal Crystalloid) ∧ (Crystalloid Equal Dobutamine) |
| 0.2113 | **\*Rule 13:**NormalUrine ← LowUrine ∧ Phenylephrine ∧ Norepinephrine ∧ (LowUrine Equal Phenylephrine) ∧ (Phenylephrine Equal Norepinephrine) |
| 0.5459 | **\*Rule 14:**NormalUrine ← LowUrine ∧ Norepinephrine ∧ Dopamine ∧ NormalArterialBE ∧ (LowUrine Equal Norepinephrine) ∧ (Norepinephrine Equal Dopamine) ∧ (Dopamine Equal NormalArterialBE) |
| 0.3926 | **\*Rule 15:**NormalUrine ← LowUrine ∧ Norepinephrine ∧ NormalRBCcount ∧ NormalBUN ∧ (LowUrine Equal Norepinephrine) ∧ (Norepinephrine Equal NormalRBCcount) ∧ (NormalRBCcount Equal NormalBUN) |
| 0.6430 | **\*Rule 16:**NormalUrine ← LowUrine ∧ Colloid ∧ NormalArterialpH ∧ Dobutamine ∧ (LowUrine Equal Colloid) ∧ (Colloid Equal NormalArterialpH) ∧ (NormalArterialpH Equal Dobutamine) |
| 0.3464 | **\*Rule 17:**NormalUrine ← LowUrine ∧ Norepinephrine ∧ Dobutamine ∧ NormalSysBP ∧ (LowUrine Equal Norepinephrine) ∧ (Norepinephrine Equal Dobutamine) ∧ (Dobutamine Equal NormalSysBP) |
| 0.5669 | **\*Rule 18:**NormalUrine ← LowUrine ∧ Phenylephrine ∧ NormalSysBP ∧ Dopamine ∧ NormalCVP ∧ (LowUrine Equal Phenylephrine) ∧ (Phenylephrine Equal NormalSysBP) ∧ (NormalSysBP Equal Dopamine) ∧ (Dopamine Equal NormalCVP) |

Table 7: Event-based temporal relations.

| Temporal Relation | Mathematical Expression | Illustration |
|---|---|---|
| $A$ Before $B$ | $t_A < t_B$ | $t_A$ ● $t_B$ ■ → |
| $A$ After $B$ | $t_A > t_B$ | $t_B$ ■ $t_A$ ● → |
| $A$ Equals $B$ | $t_A = t_B$ | $t_A$ ● $t_B$ → |

For entity $c$, this yields:
$$\mathcal{L}^c = p^c\left(t_1^c \mid \mathcal{H}_0\right) p^c\left(t_2^c \mid \mathcal{H}_{t_1^c}\right) \cdots p^c\left(t_n^c \mid \mathcal{H}_{t_{n-1}^c}\right)\left(1 - F^c\left(t \mid \mathcal{H}_{t_n^c}\right)\right) \tag{11}$$

where $p^c(t \mid \mathcal{H}_{t_n})$ represents the conditional density and $F^c(t \mid \mathcal{H}_{t_n})$ refers to its cumulative distribution function for any $t > t_n$. $\left(1 - F^c\left(t \mid \mathcal{H}_{t_n}\right)\right)$ appears in the likelihood since the unobserved point $t_{n+1}$ hasn't happened up to $t$. Further, by the hazard rate definition of the intensity function

$$\lambda^c(t) = \frac{p^c\left(t \mid \mathcal{H}_{t_n}\right)}{1 - F^c\left(t \mid \mathcal{H}_{t_n}\right)} \tag{12}$$

we will have

$$p^c(t|\mathcal{H}_{t_n}) = \lambda^c(t) \exp\left(-\int_{t_n}^t \lambda^c(s)ds\right). \tag{13}$$

Using the above equation, we can get

$$\begin{aligned}\mathcal{L}^c &= \left(\prod_{i=1}^n p^c\left(t_i^c \mid \mathcal{H}_{t_i^c-1}\right)\right) \frac{\lambda^c(t)}{p^c(t \mid \mathcal{H}_{t_n})} \\ &= \left(\prod_{i=1}^n \lambda^c(t_i^c) \exp\left(-\int_{t_{i-1}^c}^{t_i^c} \lambda^c(\tau)d\tau\right)\right) \exp\left(-\int_{t_n^c}^t \lambda^c(\tau)d\tau\right) \\ &= \left(\prod_{i=1}^n \lambda^c(t_i^c)\right) \exp\left(-\int_0^t \lambda^c(\tau)d\tau\right).\end{aligned} \tag{14}$$

Now consider the likelihood function of all entities $\mathcal{C} = \{c_1, c_2, ..., c_n\}$, which can be factorized according to the entities, the likelihood can be written as

$$Likelihood: \quad \prod_{c\in\mathcal{C}} \prod_{t_i^c\in\mathcal{H}_t} \lambda^c(t_i^c|\mathcal{H}_{t_i}) \cdot \exp\left(-\int_0^t \lambda^c(\tau|\mathcal{H}_\tau)d\tau\right) \tag{15}$$

which completes the proof.

## F  OPTIMALITY CONDITION AND COMPLEMENTARY SLACKNESS

We will provide more descriptions on the optimality condition and the complementary slackness, which provides a sound guarantee to our learning algorithm.

Given the original restricted convex problem,

$$Orignial\ Problem: \quad \boldsymbol{w}^*, b_0^* = \arg\min_{\boldsymbol{w}, b_0} -\ell(\boldsymbol{w}, b_0) + \Omega(\boldsymbol{w}); \quad s.t. \quad w_f \geq 0, \quad f \in \bar{\mathcal{F}} \tag{16}$$

where $\Omega(\boldsymbol{w})$ is a *convex* regularization function that has a high value for "complex" rule sets. For example, we can formulate $\Omega(\boldsymbol{w}) = \lambda_0 \sum_{f\in\bar{\mathcal{F}}} c_f w_f$ where $c_f$ is the rule length.

The Lagrangian of the original master problem is

$$L(\boldsymbol{w}, b_0\boldsymbol{\nu}) = -\ell(\boldsymbol{w}, b_0) + \Omega(\boldsymbol{w}) - \sum_{f\in\bar{\mathcal{F}}} \nu_f w_f, \tag{17}$$

where $\nu_f \geq 0$ is the Lagrange multiplier associated with the non-negativity constraints of $w_f$. As it is a convex problem and strong duality holds under mild conditions. Define $\boldsymbol{w}^*, b_0^*$ as the primal optimal, and $\boldsymbol{\nu}^*$ as the dual optimal, then:

$$\begin{aligned}-\ell(\boldsymbol{w}^*, b_0^*) &= \inf_{\boldsymbol{w}, b_0} L(\boldsymbol{w}, b_0, \boldsymbol{\nu}^*) \quad \text{(strong duality)} \\ &= \inf_{\boldsymbol{w}, b_0} \left(-\ell(\boldsymbol{w}, b_0) + \Omega(\boldsymbol{w}) - \sum_{f\in\bar{\mathcal{F}}} \nu_f^* w_f\right) \\ &\leq -\ell(\boldsymbol{w}^*, b_0^*) + \Omega(\boldsymbol{w}^*) - \sum_{f\in\bar{\mathcal{F}}} \nu_f^* w_f^* \\ &\leq -\ell(\boldsymbol{w}^*, b_0^*) + \Omega(\boldsymbol{w}^*).\end{aligned} \tag{18}$$

Therefore, $\sum_{f\in\bar{\mathcal{F}}} \nu_f^* w_f^* = 0$, for $f \in \bar{\mathcal{F}}$. This implies the *complementary slackness*, i.e.,

$$w_f^* = 0 \Rightarrow \nu_f^* \geq 0, \qquad w_f^* > 0 \Rightarrow \nu_f^* = 0 \tag{19}$$

Given the Karush-Kuhn-Tucker (KKT) conditions, the gradient of Lagrangian $L(\boldsymbol{w}^*, b_0^*, \boldsymbol{\nu}^*)$ w.r.t. $\boldsymbol{w}^*, b_0^*$ vanishes, i.e.,

$$\nu_f^* := -\left.\frac{\partial\left[\ell(\boldsymbol{w}, b_0) - \Omega(\boldsymbol{w})\right]}{\partial w_f}\right|_{\boldsymbol{w}^*, b_0^*}. \tag{20}$$

In summary, combining conditions (19) and (20), we obtain the optimalitiy condition of the original problem,

1. if $w_f^* > 0$, then $\nu_f^* = 0$;

2. if $w_f^* = 0$, then $\nu_f^* \geq 0$,

where the gradient $\nu_f^*$ can be computed via (20). At each iteration, we solve the subproblem to find the candidate rule that most violates this optimality condition, i.e., yields the most negative Eq. (20).

# G  ALGORITHM BOX

Our method alternates between solving a restricted master problem and a subproblem. When executing the subproblem, we need to generate several candidate rules. We summarize the algorithm in Algorithm 1, Algorithm 2, and Algorithm 3. RG refers to the Rule Generator used to generate a new candidate rule when solving the subproblem. Here, we will parameterize the RG as a LSTM. SP is the abbreviation of Sub-Problem which is optimized to construct a new rule. RMP indicates the Restricted Master Problem used to update model parameters.

---

**Algorithm 1:** Rule Generator (RG)

---

**Input:** RuleLen, HeadPred
**Output:** CandidateRule

1
2  PredLibrary ← $\{A, B, \dots\}$;
3  TempRelationLibrary← $\{Before, After, Equal, None\}$;
4
5  Sequence ← EmptySet;
6  BodyPredSet ← EmptySet;
7
8  DynamicMask = All-Ones
9
10 **while** *BodyPredNum≤ RuleLen* **do**
11 $\quad$ NewBodyPred ← LSTM(Sequence, PredLibrary ⊙ DynamicMask)
12 $\quad$ DynamicMask ← Set the location of NewBodyPred zero (DynamicMask)
13 $\quad$ Sequence.add(NewBodyPred);
14
15 $\quad$ **for** *BodyPred in BodyPredSet* **do**
   $\quad\quad$ // We need to consider the pairwise temporal relation
   $\quad\quad\quad$ between the new selected body predicate and all the
   $\quad\quad\quad$ body predicates that already have been selected in the
   $\quad\quad\quad$ body predicate set.
16 $\quad\quad$ TempRelation ← LSTM(BodyPred, NewBodyPred, TempRelationLibrary);
17 $\quad\quad$ Sequence.add(TempRelation);
18
19 $\quad$ BodyPredSet.add(NewBodyPred);
20
21 CandicateRule ← Convertor(Sequence); // Convert a sequence of tokens to
   $\quad$ rule template.
22
23 **return** *CandicateRule*

---

---

**Algorithm 2:** Sub-Problem (SP)

---

**Input:** RuleLen, HeadPred
**Output:** NewRule

---

1
2 **while** *iter ≤ Total_Iter* **do**
3      CandidateRuleBatch ← EmptySet
4
5      **while** *Batch_idx ≤ BatchSize* **do**
6          CandidateRule ← RG(RuleLen, HeadPred); // Generate candidate rule.
7          CandidateRuleBatch.add(CandidateRule);
8
9      Policy ← Policy.update// Policy gradient.
10
11      **if** *PolicyGradientNorm ≤ threshold* **then**
12          FinalCandidateRule ← RG(RuleLen, HeadPred);
13          **return** *FinalCandidateRule*
14 FinalCandidateRule ← RG(RuleLen, HeadPred);
15 **return** *FinalCandidateRule*

---

**Algorithm 3:** Complete model

---

**Input:** RuleLen, HeadPred, TotalRuleNum
**Output:** ruleSet

---

1
2 ruleSet ← EmptySet;
3 $b \leftarrow 0$;
4 $w \leftarrow 0$;
5 $b, w \leftarrow$ RMP($b, w$, ruleSet); // Initialize weights and base terms.
6
7 **while** *CurrentRuleNum ≤ TotalRuleNum* **do**
8      NewRule ← SP(RuleLen, HeadPred); // Generate candidate rule.
9
10      **if** *IncreasedGain<0* **then**
11          ruleSet.add(NewRule);
12          $b, w \leftarrow$ RMP($b, w$, ruleSet); // After adding new rule, update
                weights and base terms.
13 **return** $b$*, $w$, ruleSet*

---

## H COMPUTING INFRASTRUCTURE.

All experiments are performed on Ubuntu 20.04 LTS system with Intel(R) Xeon(R) CPU E5-2690 v3 @ 2.60GHz CPU, 112 Gigabyte memory and single NVIDIA Tesla P100 accelerator.

## I COMPLETE RESULTS OF EXPERIMENTS ON SYNTHETIC DATA

Fig. 7 and Fig. 8 demonstrate the learning results for the cases with predicate size 8, 12, 16, 20, and 24 (predicates in the library that we need to search) for all groups using 1000 samples of networked events. We set the learning rate in solving the subproblem to be $\times 10^{-2}$. Hidden state size of LSTM was 32. The learning rate in solving the restricted master problem was $\times 10^{-3}$. Our proposed model discovered almost all the ground truth rules for all cases in all groups. And almost all the truth rule weights are accurately learned and the MAEs are quite low. For cases in group-3 and group-6, we considered long and complex rules with 8 body property predicates and the associated temporal relations, which yield a very big search space, but our model still discovered most of the ground truth rules.

We also evaluated our model using 500 and 2000 samples with the same experiment settings. Our model also achieved satisfactory performance when sample size is small, and the performance of the model was further improved as the sample size increases.

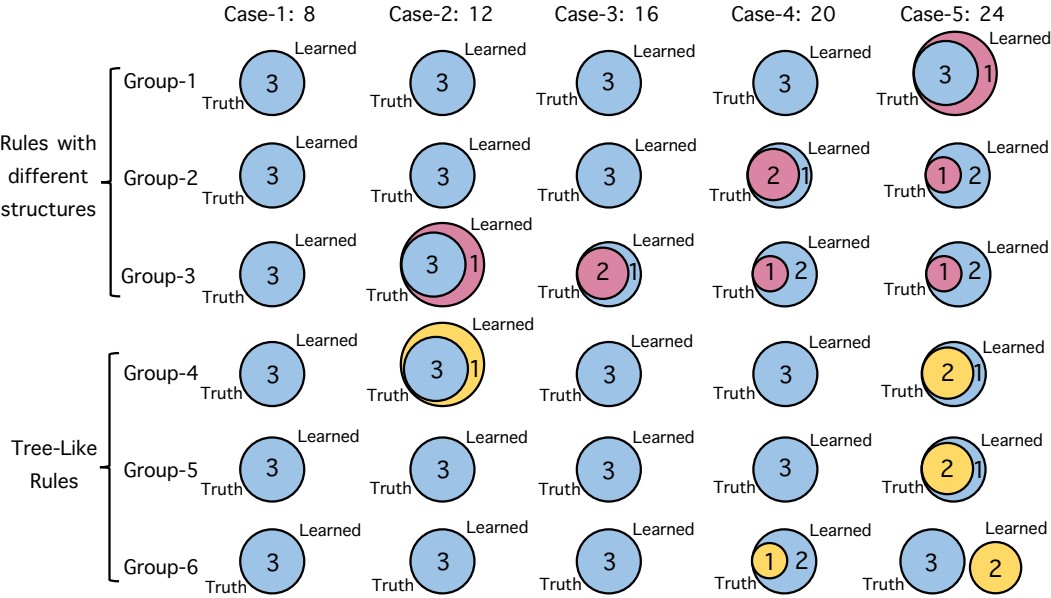

Figure 7: Jaccard similarity score for all 6 groups and all 5 cases using 1000 samples. Blue one indicates the ground truth rule and red/yellow one indicates the learned rule in different groups.

## J    REUSE THE LSTM MEMORY AND EARLY STOP MECHANISM

**When to reuse the LSTM memory for different subproblem iterations?**    The ground truth rules in group-$\{1, 2, 3\}$ are quite distinct in their content with almost no common predicates in each rule. Given this prior knowledge, when training the datasets in these groups, whenever the algorithm enters the subproblem to search for a new candidate rule, we reset our LSTM model and clean the memory. If not, we will get the convergence results as illustrated in Fig. 9, where we observe that keeping the LSTM memory will hinder the convergence speed of the subproblems. As illustrated in Fig. 10 (left) where we refresh the LSTM model parameters whenever the algorithm enters the subproblem stage. As a comparison, by doing this, the convergence of the subproblem is much faster. The ground truth rules in group-$\{4, 5, 6\}$ are similar in content and share many predicates. Given this prior knowledge, reuse the LSTM across subproblems may help the convergence.

**Early stop mechanism.**    To speed up our algorithm, we propose an early stop mechanism. We consider that when the iteration times reach a pre-set reasonable number, and when the LSTM model consecutively generates identical logic rules (like identical 10 rules), we conclude that the LSTM model has been trained enough and is able to generate a good-performing logic rule. We don't need to train the LSTM until the norm of policy gradient is within a very small tolerance. Hence, we may early stop our training. Fig. 10 (right) illustrates the training trajectories of solving subproblem for group-1 case-1, where we reset the LSTM in the beginning of the subproblems and use early stop mechanism. It's obvious that this mechanism can reduce redundant iterations and help us complete the training process in a short time.

## K    COMPARE WITH STANDARD POLICY GRADIENT

For standard policy gradient, the policy $\pi_\theta(a|s)$ is trained end-to-end by minimizing the subproblem objective using policy gradient, i.e.,

$$\max_\theta J(\theta) = \max_\theta \mathbb{E}_{\tau \sim \pi_\theta(\tau)}[R(\tau)]$$

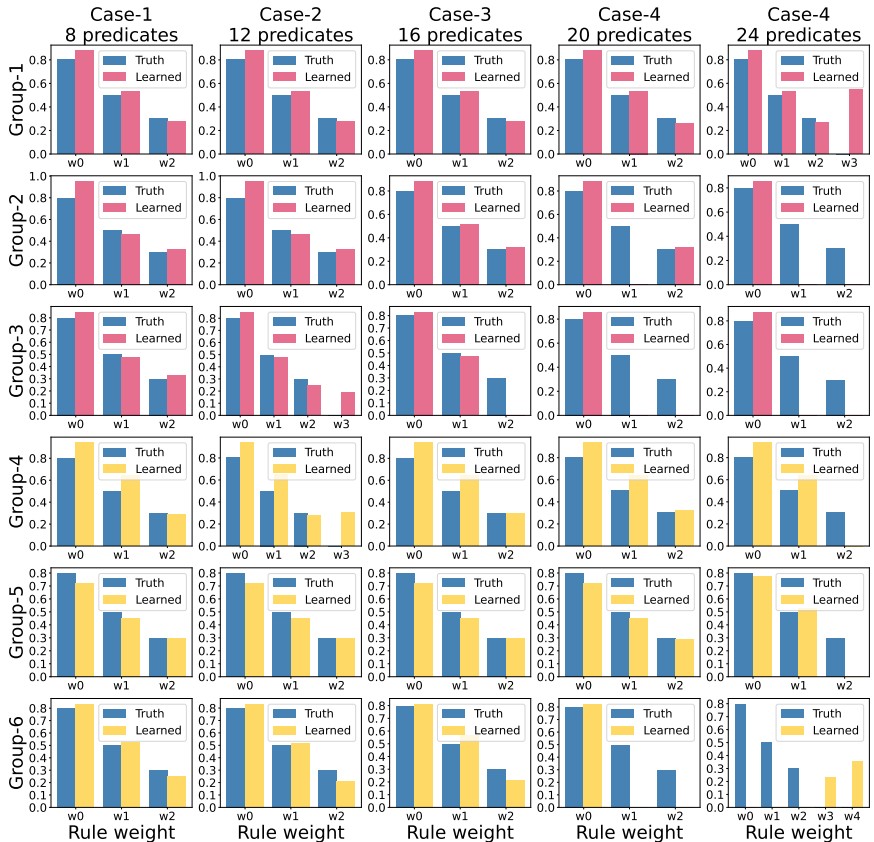

Figure 8: MAE values for all 6 groups and all 5 cases using 1000 samples. Blue one indicates the ground truth rule and red/yellow one indicates the learned rules in different groups.

where $\tau$ is the generated tokens (i.e., candidate rule) and $R(\tau) = \frac{\partial \ell(\boldsymbol{w}, b_0)}{\partial w_f} - \frac{\partial \Omega(\boldsymbol{w})}{\partial w_f}\Big|_{\boldsymbol{w}_{(k)}^*, b_{0,(k)}^*}$. The policy gradient can be estimated using the roll-out samples, i.e.,

$$\nabla_\theta J(\theta) \approx \frac{1}{N} \sum_{i=1}^{N} \sum_{k=1}^{K} \nabla_\theta \log \pi_\theta(a_k^{(i)} | s_k^{(i)}) R^{(i)}(\tau) \tag{21}$$

where $N$ is the number of episodes, and $K$ is the length of tokens (actions). We use this estimated policy gradient to update the policy $\theta$, $\theta \leftarrow \theta + \alpha \nabla_\theta J(\theta)$, where $\alpha$ is the learning rate.

However, in our problem, the final performance of our model is measured by the single or few best-performing rules found during training. So normal policy gradient may not be satisfactory while risk-seeking policy gradient may be more suitable.

We compare the performance of the risk-seeking policy gradient model with the normal policy gradient model for cases in group-3 and group-6, mainly because the ground truth rules in these two groups are long and complex with many body property predicates and various temporal relations. Due to the difficulty of recovering ground truth rules in these groups, it may be more obvious to distinguish the performance of the two models. We set $\epsilon$ to be 0.3. The learning rate in solving the subproblem was $\times 10^{-2}$ and the hidden state size of LSTM was 32. The learning rate in solving the restricted master problem was $\times 10^{-3}$. The results are shown in Fig. 11 and Fig. 12.

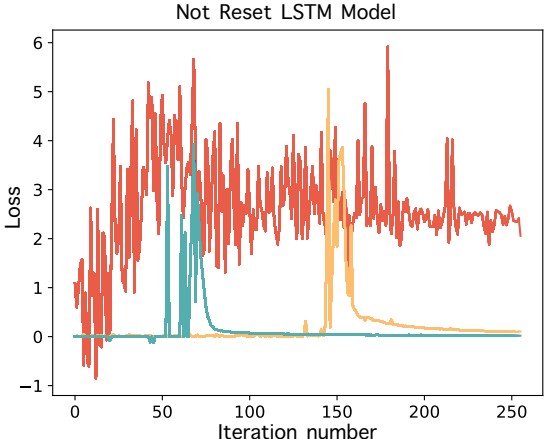

Figure 9: Training loss trajectories of solving different subproblem stages if the LSTM model parameters are inherited from the previous subproblem.

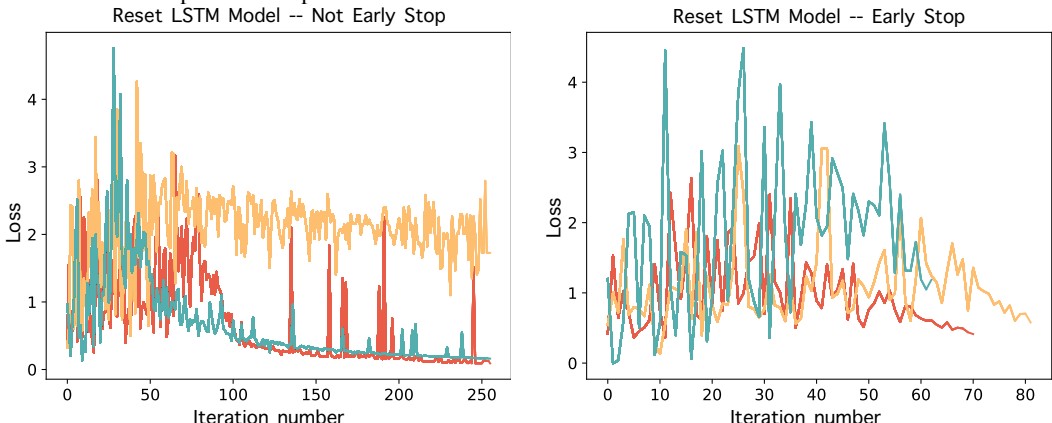

Figure 10: Training loss trajectories of solving different subproblem stages for group-1 case-1. Left: The LSTM parameters are refreshed in the beginning of each subproblem. Right: Early stop mechanism is used.

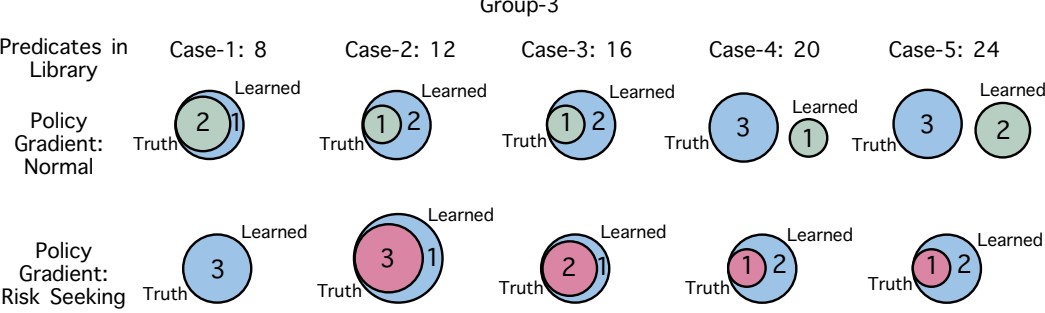

Figure 11: Jaccard similarity score for all 5 cases in group-3 using 1000 samples. Blue one indicates the ground truth rule, green one indicates the learned rule using normal policy gradient, and red one indicates the learned rule using risk-seeking policy gradient.

By using risk-seeking policy gradient, there are actually some significant improvements. The results show that more, and more importantly, more accurate ground truth rules were learned by the risk-seeking policy gradient.

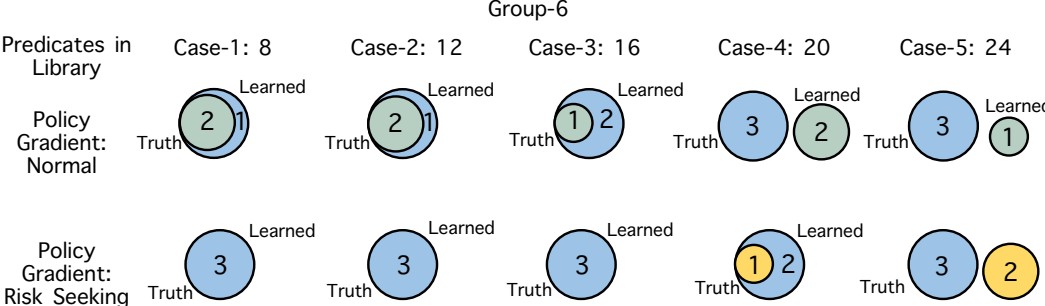

Figure 12: Jaccard similarity score for all 5 cases in group-6 using 1000 samples. Blue one indicates the ground truth rule, green one indicates the learned rule using normal policy gradient, and yellow one indicates the learned rule using risk-seeking policy gradient.

## L    PREDICATE DEFINITION IN MIMIC-III

MIMIC-III is an electronic health record ICU dataset, which is released under the PhysioNet Credentialed Health Data License 1.5.0[1]. It was approved by the Institutional Review Boards of Beth Israel Deaconess Medical Center (Boston, MA) and the Massachusetts Institute of Technology (Cambridge, MA). In this dataset, all the patient health information was deidentified. We manually checked that this data do not contain personally identifiable information or offensive content.

We defined 63 predicates, including two groups of drugs (i.e., intravenous fluids and vasopressors) and lab measurements, see Tab. 8 for more details. Among all these predicates we were interested in reasoning about two predicates and define them as head predicates: 1) LowUrine and 2) NormalUrine. We treated real time urine as head predicates since low urine is the direct indicator of bad circulatory systems and the signal for septic shock; normal urine reflects the effect of the drugs and treatments and the improvement of the patients physical condition. In our experiments, lab measurement variables were converted to binary values (according to the normal range used in medicine) with the transition time recorded. For drug predicates, they were recorded as 1 when they were applied to patient. We extracted 2023 patient sequences, and randomly selected 80% of them for training and the remaining 20% for testing. The average time horizon is 392.69 hours and the average events per sequence is 79.03.

## M    EXPERTS' MODIFICATION ALSO IMPORTANT

We also invite human experts (i.e. doctors in ICU) to justify correctness and modify our learned rules. We compare the log-likelihood trajectories of directly updating the rule weights and directly updating the weights of the modified rules by human experts, and the results are shown in Fig. 13. The results show that the modification suggestion from human experts can indeed help improve the performance of our model, since the log-likelihood trajectory of modified rules rise higher and faster.

## N    REAL-WORLD EXPERIMENT: EYE FIXATION

Simple choices are made by integrating noisy evidence that is sampled over time and influenced by visual attention. As a result, fluctuations in visual attention can affect choices. We aim to understand shoppers' purchase patterns given their eye fixation event data (Callaway et al., 2021).

Our conjecture is: the location of the items, shopper-assessed values of the items, and the shopper's visual habits (usually looking from left to right) will affect their final item choice. We learned temporal logic rules and their weights to quantitatively understand this.

**Dataset Description:** Three items randomly placed on the "left", "middle" and "right" on the supermarket shelf, each has a unique "price" (value).

Each shopper evaluated three items by eye fixation until they identified an item to purchase.

---

[1]https://physionet.org/content/mimiciii/view-license/1.4/

Table 8: Defined Predicates in Our MIMIC-III Experiment.

| | |
|---|---|
| Lab Measurements | `Low/Normal/High-SysBP`
`Low/Normal/High-SpO2SaO2`
`Low/Normal/High-CVP`
`Low/Normal/High-SVR`
`Low/Normal/High-Potassium`
`Low/Normal/High-Sodium`
`Low/Normal/High-Chloride`
`Low/Normal/High-BUN`
`Low/Normal/High-Creatinine`
`Low/Normal/High-CRP`
`Low/Normal/High-RBCcount`
`Low/Normal/High-WBCcount`
`Low/Normal/High-ArterialpH`
`Low/Normal/High-ArterialBE`
`Low/Normal/High-ArterialLactate`
`Low/Normal/High-HCO3`
`Low/Normal/High-SvO2ScvO2` |
| Output | `LowUrine`, `NormalUrine` |
| Input | `Colloid`, `Crystalloid`, `Water` |
| Drugs | `Norepinephrine`, `Epinephrine`,
`Dobutamine`, `Dopamine`, `Phenylephrine` |
| Temporal Relation Type | `Before`, `After`, `Equal` |

Figure 13: Log-likelihood trajectory for just running master problem to update rule weights. Blue curve indicates that we put all learned rules directly into master problem and update their weights. Red curve indicates that we put the modified rules directly into master problem and update their weights.

The data record each shopper's eye fixated items, when and where, and their final purchased item. There are 30 participants, each with at most 100 independent trials. At each trial, participants were asked to look at these three items and choose the item that they think is most valuable. There are 2966 trials in total. On average, for one trial, a participant has 4.3011 eye fixations.

**Predicate Definition:** We are interested in explaining three final choices of a shopper:

1) finally choose the item with the actual (not shopper-assessed) largest value

2) finally choose the item with the last eye fixation, and

3) finally choose the item with the longest eye fixation.

These final choices define the head predicate set. Another 18 predicates are about the location of the items, value of the items, and the time of eye fixation of a shopper on one specific item. Please refer the Tab. 9 below for a complete predicate set.

Table 9: Defined predicates for eye fixation trials. For eye fixation predicates, there are three parts: location of item, value of item, duration of eye fixation

| | |
|---|---|
| Eye Fixation | `Left_MaxValue_LongFixation` |
| | `Left_MaxValue_ShortFixation` |
| | `Left_MidValue_LongFixation` |
| | `Left_MidValue_ShortFixation` |
| | `Left_MinValue_LongFixation` |
| | `Left_MinValue_ShortFixation` |
| | `Middle_MaxValue_LongFixation` |
| | `Middle_MaxValue_ShortFixation` |
| | `Middle_MidValue_LongFixation` |
| | `Middle_MidValue_ShortFixation` |
| | `Middle_MinValue_LongFixation` |
| | `Middle_MinValue_ShortFixation` |
| | `Right_MaxValue_LongFixation` |
| | `Right_MaxValue_ShortFixation` |
| | `Right_MidValue_LongFixation` |
| | `Right_MidValue_ShortFixation` |
| | `Right_MinValue_LongFixation` |
| | `Right_MinValue_ShortFixation` |
| Final Choice | `FinalChoice_LargestValue` |
| | `FinalChoice_LastFixation` |
| | `FinalChoice_LongestFixation` |
| Temporal Relation Type | `Before` |

**Rules Discussion:** We displayed the discovered important rules in Tab. 10, which summarize the eye fixation patterns before shoppers making choices. From the results, we have the following discoveries:

1) the final fixation is shorter

2) the later (but not the final) fixations are longer

3) people are more likely to begin to look from the left or from the middle.

Specifically, if a shopper finally chooses the item with the largest value, he may first glance over all three items at least once, or after looking at all three items, go back to check the item he wants to choose, and then make a choice (Rule 1, 2, 5, and 8).

And people are more used to looking from left to right (Rule 2, 3, 5, and 7).

If a shopper finally chooses the item with the last eye fixation, he may only take a quick look at these items and may miss one or two of these items (Rule 3, 6, and 9). If a shopper finally chooses the item with the longest eye fixation, he may spend a lot of time on most of these items, reevaluating the value of these items back and forth in his mind (Rule 4, 7, and 10).

In summary, our discovered temporal logic rules provide insight into shopper's perchance behaviors in terms of eye fixation patterns.

Table 10: Temporal Logic Rules Discovered for Eye Fixation and Final Choice. Since there is only one type of temporal relation "Before", in the following representation of temporal logic rules, we just ignore the temporal relation

| Weight | Rule |
|---|---|
| 0.0550 | **Rule 1:** FinalChoice_LargestValue ← Middle_MaxValue_LongFixation $\wedge$ Left_MidValue_ShortFixation $\wedge$ Right_MinValue_ShortFixation |
| 0.0976 | **Rule 2:** FinalChoice_LargestValue ← Left_MaxValue_LongFixation $\wedge$ Middle_MinValue_ShortFixation $\wedge$ Right_MidValue_ShortFixation |
| 0.0278 | **Rule 3:** FinalChoice_LastFixation ← Left_MaxValue_LongFixation $\wedge$ Middle_MinValue_ShortFixation $\wedge$ Left_MaxValue_ShortFixation |
| 0.0479 | **Rule 4:** FinalChoice_LongestFixation ← Left_MaxValue_LongFixation $\wedge$ Middle_MidValue_LongFixation $\wedge$ Left_MaxValue_ShortFixation |
| 0.0648 | **Rule 5:** FinalChoice_LargestValue ← Left_MaxValue_LongFixation $\wedge$ Middle_MidValue_ShortFixation $\wedge$ Right_MinValue_LongFixation $\wedge$ Left_MaxValue_ShortFixation |
| 0.0015 | **Rule 6:** FinalChoice_LastFixation ← Middle_MidValue_ShortFixation $\wedge$ Left_MaxValue_LongFixation $\wedge$ Middle_MidValue_ShortFixation $\wedge$ Left_MaxValue_ShortFixation |
| 0.0457 | **Rule 7:** FinalChoice_LongestFixation ← Left_MinValue_LongFixation $\wedge$ Middle_MaxValue_LongFixation $\wedge$ Right_MidValue_ShortFixation $\wedge$ Middle_MaxValue_ShortFixation |
| 0.0241 | **Rule 8:** FinalChoice_LargestValue ← Middle_MidValue_ShortFixation $\wedge$ Left_MaxValue_LongFixation $\wedge$ Right_MinValue_ShortFixation $\wedge$ Middle_MidValue_LongFixation $\wedge$ Left_MaxValue_ShortFixation |
| 0.0243 | **Rule 9:** FinalChoice_LastFixation ← Middle_MidValue_ShortFixation $\wedge$ Right_MinValue_ShortFixation $\wedge$ Left_MaxValue_LongFixation $\wedge$ Middle_MidValue_LongFixation $\wedge$ Left_MaxValue_ShortFixation |
| 0.0145 | **Rule 10:** FinalChoice_LongestFixation ← Left_MaxValue_LongFixation $\wedge$ Middle_MidValue_ShortFixation $\wedge$ Left_MaxValue_LongFixation $\wedge$ Right_MinValue_LongFixation $\wedge$ Left_MaxValue_ShortFixation |

