# OpenReview forum: "Reinforcement Logic Rule Learning for Temporal Point Processes "
_ICLR.cc/2023/Conference — Submitted to ICLR 2023_

### Official Review · Reviewer_Snuq · 2022-10-24

**Confidence:** 3
**Correctness:** 2
**Technical Novelty And Significance:** 3
**Empirical Novelty And Significance:** 3
**Recommendation:** 6

**Clarity, Quality, Novelty And Reproducibility:**

This work notes that enumeration is intractable for long temporal logic rules.
Tractability of their methods was successfully demonstrated.

Section 5.1 would have benefited from results for additional methods within the main body for comparison.
Likewise, the baselines mentioned in 5.2 are not used for earlier comparisons.

Minor comments:
- Several times, the authors use a phrase akin to "We make the temporal logic rule search subproblem differentiable." This is not entirely true. The RL policy is differentiable, but the search problem itself is not differentiable. Only the problem of finding logic rules _with this given policy_ is differentiable.
- Parenthetical and non-parenthetical citations are each sometimes used when the other would be more appropriate. Another editing pass for this would be helpful.
- 4.2.2: "wihch" -> "which"
- 5.2: "studies suggests" -> "studies suggest"

The results in Table 1 (OURMETHOD) are prior to doctor intervention, right?
What is the performance after the doctors have made modifications?

**Strength And Weaknesses:**

The method is an intuitive improvement over the existing enumeration approach.
My understanding of the method is that the selection of rules is done in a greedy way.
Additionally, further rules are not added when the next greedy rule is no longer useful.
As a result, this method is akin to performing prioritized enumeration along with a stopping condition.

My understanding is that the state only includes information about the tokens for the current rule (i.e., no information is included about existing rules).
As a result, the typical MDP properties do not hold in this setting.
The choice of an RL formulation is therefore a bit odd, but the sequential nature does make it a better fit than alternatives with which I am familiar.

This work seeks to also perform transfer. Transfer in RL is a non-trivial problem.
The assumption of "same or similar ground truth rules" substantially simplifies the problem. This assumption is akin to assuming similar optimal policies, at which point policy reuse (without adaptation) is expected to substantially improve early performance.
Benefiting from past knowledge when able while not hurting performance in other cases is difficult. This aspect is stripped away in the performed transfer experiments as transfer is known to be helpful.
Additionally, this assumption cannot be made in real-world applications of this method as ground truth rules cannot be known prior to applying methods.

Section 5.2 with human experts has experts looking at rules and determining if these rules seem reasonable.
This approach to evaluation is only considering one side: do doctors generally agree with the rules identified.
The other side is missing: do doctors disagree with the rules not selected by the system.
Since the system is supposed to distinguish between useful and non-useful rules, this categorization should be evaluated by doctors.
Ideally, this should be done close to the decision boundary, so comparisons should be made between the last included rules and the first ones not included (rather than comparing random rules to the top rules, for example).
Additionally, the goal of explanations is to provide trust in _performant_ systems.
Evaluation can be repeated with rules generated by a faulty model, and doctors should be able to distinguish between the properly functioning and faulty models based on the produced explanations.

**Summary Of The Paper:**

This work presents a method for identifying temporal logic rules.
This method uses reinforcement learning to select rules and separately optimizes the weights of these rules.

This method is evaluated on finding rules on benchmark tasks, in a transfer setting, and on a real-world problem.

**Summary Of The Review:**

The new method is motivated well and appears to perform better than existing alternatives.
Evaluation could be improved, though. For example:
For the real-world study, by only considering the useful rules found by a successful system, the usefulness of explanations is not evaluated.

---

> ### Author Response · Authors · 2022-11-18
> **In response to reviewer Snuq**
>
> Thank you for the careful reading and constructive comments. Your suggestions will help us continue improving the overall quality of our paper. In the following, we will address each of your concerns in detail.
>
> $\star\star\star \textbf{Do doctors disagree with the rules not selected by the system?}$
>
> Yes, doctors disagree with the rules not selected by the system. In our experiments, there is a pre-set threshold for rule weight. Learned rules with weights below this threshold would be discarded during the alternating process between master problem and subproblem of our proposed algorithm. To name a few discarded rules:
>
> ```
> ----------------------------------------------------------------------------------
> Rule-1: LowUrine <-- LowSysBP ^ HighBUN ^ LowArterialpH ^
>                        (LowSysBP Before HighBUN) ^ (LowSysBP After LowArterialpH) ^
>                        (HighBUN Before LowArterialpH)
> ----------------------------------------------------------------------------------
> Rule-2, weight=0.0006: LowUrine <-- HighSysBP ^ HighBUN ^ LowBUN ^ LowSVR ^
>                        (HighSysBP Before HighBUN) ^ (HighBUN Equal LowBUN) ^
>                        (LowBUN Before LowSVR)
> ----------------------------------------------------------------------------------
> ```
>
> Verified by doctors, Rule-1 has inherently wrong temporal logic relation, since low systolic blood pressure of a patient (LowSysBP) happened before high blood urea nitrogen of a patient (HighBUN), LowSysBP happened after low arterial blood ph of a patient (LowArterialpH), but HighBUN happened before LowArterialpH, which is impossible.
>
> And doctors also disagree with Rule-2 and consider it as clinical meaningless, since LowSysBP is not an indicator of bad circulatory systems and the signal for septic shock, which would be reflected by the patient's low urine output. Moreover, HighBUN would never happen at the same time as LowBUN.
>
> $\star\star\star \textbf{Regarding more details about the results in Table 1 (OURMETHOD) when introducing doctor intervention}$
>
> Yes, the results in Table 1 (OURMETHOD) are prior to doctor intervention, the MAE is slightly decrease to 1.612 and 1.298 for LowUrine and NormalUrine respectively after the doctors have made modifications. And we also compared the log-likelihood trajectories of directly updating the rule weights and directly updating the weights of the modified rules by human experts, and the results are shown in Fig.13 in Appendix M in our paper. The results show that the modification suggestion from human experts can indeed help improve the performance of our model, since the log-likelihood trajectory of modified rules rises higher and faster.
>
> $\star\star\star \textbf{Regrading the sentence which is not rigorous}$
>
> In Section 1 of our paper -- last paragraph above the contributions – we say that “1) We make the temporal logic rule search subproblem differentiable’’.  We agree with you that this is not entirely true and indeed only the RL policy is differentiable. Actually, we aim to illustrate that our subproblem is a differentiable policy-guided rule search algorithm to learn explanatory temporal logic rules, and all the policy parameters can be learned end-to-end via policy gradient using the subproblem objective as reward. To make our writing more rigorous, we have changed this sentence to “We utilize differentiable policy gradient to solve the temporal logic rule search subproblem” in the revised paper.

---

> ### Author Response · Authors · 2022-11-18
> **In response to reviewer Snuq**
>
> $\star\star\star \textbf{Doctors should be able to distinguish between the properly functioning and faulty models based on the produced explanations.}$
>
> This suggestion is very constructive! We did this suggested experiment in the last few days. As a domain knowledge, blood potassium level is highly related to the symptom of sepsis. High blood potassium level indicates that patients may suffer from typical symptoms of sepsis, such as low urine output. Normal range of blood potassium level is 3.7 to 5.2 milliequivalents per liter (mEq/L). For a properly functioning model, we define blood potassium level greater than 5.2 mEq/L as high blood potassium level. For a faulty model, we wrongly define blood potassium level greater than 5.2 mEq/L as low blood potassium level.
>
> Setting same random seed for both properly functioning model and faulty model, properly functioning model would yield a learned rule:
>
> ```
> ----------------------------------------------------------------------------------
> Rule-1, weight=0.4935: LowUrine <-- LowSysBP ^ HighArterialLactate ^ HighPotassium ^
>                        HighChloride ^ HighBun ^ (LowSysBP Equal HighArterialLactate) ^
>                        (HighArterialLactate Equal HighPotassium) ^
>                        (HighPotassium Equal HighChloride) ^
>                        (HighChloride Equal HighBun)
> ----------------------------------------------------------------------------------
> ```
>
> where “HighPotassium” indicates high blood potassium level. This temporal logic rule is verified by the doctor and the doctor confirmed that this rule is clinically meaningful.
>
> But a faulty model would yield a learned rule which is obviously not clinically significant:
>
> ```
> ----------------------------------------------------------------------------------
> Rule-2, weight=0.4935: LowUrine <-- LowSysBP ^ HighArterialLactate ^ LowPotassium ^
>                        HighChloride ^ HighBun ^ (LowSysBP Equal HighArterialLactate) ^
>                        (HighArterialLactate Equal LowPotassium) ^
>                        (LowPotassium Equal HighChloride) ^
>                        (HighChloride Equal HighBun)
> ----------------------------------------------------------------------------------
> ```
>
> where “LowPotassium” would not be a signal for septic shock. Only by analyzing such obviously wrong rules, doctors can judge that there may be issues with our model, and the faulty model may yield such clearly clinical meaningless temporal logic rules.
>
> The above simple example is just a wrong model at the data level. We can also construct the wrong model by multiplying the objective function of the subproblem by $-1$, so that the direction of the optimization will be completely opposite, which will also produce many counterintuitive and clinical meaningless temporal logic rules. Therefore, it is easy for doctors to distinguish temporal logic rules with obvious errors, and it is also easy to judge whether our model is reliable or not, just based on the learned temporal logical rule.
>
> To sum up, our proposed method makes doctors easily distinguish between the properly functioning and faulty models based on the produced explanations.

---

### Official Review · Reviewer_Z32A · 2022-10-25

**Confidence:** 2
**Correctness:** 2
**Technical Novelty And Significance:** 2
**Empirical Novelty And Significance:** 2
**Recommendation:** 3

**Clarity, Quality, Novelty And Reproducibility:**

In general, I found that many statements in the paper are either inaccurate or poorly justified:
- Section 1 - last paragraph above the contributions -- "1) We make the temporal logic rule search subproblem differentiable" -- I am not sure if this statement is accurate. The sentence seems to imply that there are gradients that can be computed through the search procedure, but I think only policy gradient is computed for optimization as far as what I understand.
- Why does $w_f$ need to be positive in Equation (4, 5)?
- Section 4.1 - "For example, we can formulate $\Omega(\mathbf{w}) = \lambda\_0 \sum\_{f \in \bar{\mathcal{F}}} c_f w_f$ where $c_f$ is the rule length". I am confused by this regularization function. If $w_f$ is a vector of parameters and $c_f$ is a scalar, how can the multiplication results in a scalar?
- Should the minimization be over $f$ rather than $\phi\_f$ in Equation (7)? It does not make much sense to minimize the logic-informed function as the function itself (from my understand) remains fixed (when conditioned on $f$). Although I could be misunderstanding it as it was not very clear in the setup whether this feature function is fixed.
- Section 5.1 accuracy and scalability - "1) whether our reinforcement temporal
logic learning algorithm can truly uncover the rules from the noisy variable set" -- what is the noisy variable set referring to? How is the noise being introduced?

There are also many typos and grammar issues through out the paper that makes the paper hard to read and understand. To name a few:
- Page 3 above Equation 1 -- "By some simple proof Rasmussen (2018)"
- Page 4 Section 3.2 Temporal Logic Rule -- "Add a temporal dimension to the predicates.": incomplete sentence.
- Page 4 Section 3.3 Temporal Logical Point Process -- "Introduce a logic function $g_f(\cdot)$ to check ...": incomplete sentence. Also, is $g_f(\cdot)$ a learnable function? What does it output?

Many notations are not introduced properly:
- What is $X_u(t_u)$ in Equation (2)?
- How is $g_f(\cdot)$ defined in Equation (3)? The paper mentions that it is a logic function that checks the body conditions of $f$, but this is very vague and it is unclear what the output of the function really is.


**Strength And Weaknesses:**

*Strength*

- The idea of formulating the logic rule generation problem as a reinforcement learning problem is interesting.

*Weaknesses*

- The paper has a lot of incorrect statements and confusing sentences which make the technical content hard to understand.
- The empirical section seems to only contain results generated using one single run of the algorithm, which makes it very difficult to judge the effectiveness of the proposed algorithm.
- Many details of the baseline algorithms are missing in Table 1 (e.g., Transformer architecture, network size, how the data are fed into the transformer models) which could pose major reproducibility issues.

**Summary Of The Paper:**

The paper proposes to learn a set of temporal logic rules to build a probabilistic model that maximizes the likelihood of the observed data that come in the form of temporal event sequences. Compared to brute-force search in the space of logic rules that are enormous, the proposed approach leverages a neural policy to generate logic rules guided by heuristic reward signals to maximize the likelihood of the probabilistic model that can be obtained using the new rules. Empirically, the paper demonstrates that the proposed approach can improve upon prior baselines that utilize brute force for logic rule generations on a set of synthetic event data proposed by the authors. The proposed approach also outperforms prior methods on the MIMIC-III health care dataset by achieving lower prediction errors.

**Summary Of The Review:**

In general, I found the paper to be a bit difficult to understand. I can roughly get the main idea of the propose method, but I might have misunderstood many technical details due to the writing issues. In addition, I found that the empirical evaluation of the paper to be a bit lacking and the details of some baselines are missing. These altogether make it hard to evaluate the significance of the results. Because of these reasons, I would not recommend acceptance of the paper at its current state.

---

> ### Author Response · Authors · 2022-11-16
> **In response to reviewer Z32A**
>
> Thank you for spending time reading our paper and providing your comments!
>
> $\star\star\star \textbf{Regrading the sentence which is not rigorous}$
>
> In Section 1 of our paper -- last paragraph above the contributions – we say that “1) We make the temporal logic rule search subproblem differentiable’’. What we want to emphasize in this sentence is that our subproblem is a differentiable policy-guided rule search algorithm, and all the policy parameters can be learned end-to-end via policy gradient using the subproblem objective as reward. This is very different from other enumeration methods or black-box optimization methods such as generic algorithm, which are non-differentiable. To avoid your confusion, we have changed this sentence to “We utilize differentiable policy gradient to solve the temporal logic rule search subproblem” in the revised paper.
>
> $\star\star\star \textbf{Regarding Equation (4) and (5) in our paper}$
>
> $w_f$ is constrained to be positive because given our model assumption, a positive rule weight makes more sense. In our model, the logic-informed feature $\phi_f$ could be either positive, when the head predicate has a positive sign in the rule,  or negative, when the head predicate has a negation sign in the rule. Negative features indicate that when these rules hold, the occurrence of the head predicate event will be suppressed due to the negation sign.  Assuming positive rule weight is more reasonable. What’s more, the positive weight can be interpreted as the importance score of a logic rule, which is more intuitive.
>
> $\star\star\star \textbf{Regarding the regularization function in Section 4.1}$
>
> Sorry that you don’t understand our notation. Let’s clarify again. According to our notation, $w_f$ indicates the rule weight of a “specific” rule $f$ and $c_f$ is the length of this rule. Rule $f$ is one specific rule that is included in the rule set $F$. So both $w_f$ and $c_f$ are scalars. It has no problems in the regularization term.
>
> $\star\star\star \textbf{Regarding the question about minimization over $f$ or $\phi_f$}$
>
> Minimization over $f$ and $\phi_f$ in Equation (7) in our paper are equivalent in our setting. To avoid your confusion, we will change it to $f$.
>
> $\star\star\star \textbf{Regarding the noise predicates in Section 5.1}$
>
> Sorry that you don’t get the meaning of the noise predicates in our experiments. Noise predicate means redundant predicate, which is not the component of any ground-truth rules. Noise predicates serve like background noise. More noise predicates will make the rule learning problem more challenging.
>
> $\star\star\star \textbf{Regarding the issues of notations}$
>
> (1) $X_u (t_u)$ in Equation (2) in our paper indicates the body property predicate happens at time $t_u$. $u \in X_f$  indicates that predicate $u$ belongs to the set of predicates defined in rule $f$, namely $X_f$. Optionally, we can define it as $X_u \in X_f$, where $X_u$ belongs to the set of predicates defined in rule $f$.
>
> (2) The function $g_f (\cdot)$ is not a learnable function in our paper and experiments (although it can be learnable). To avoid your confusion, we have elaborated more on this part in our paper.
>
> We have carefully thought about all the issues you mentioned and made corresponding responses and revisions. We wonder whether our response has addressed your major concerns about writing and notations to allow the paper to cross your acceptance threshold. If you have any additional specific concerns, please let us know, and we would be happy to answer them or address them in the final version.

---

### Official Review · Reviewer_XGxF · 2022-10-25

**Confidence:** 3
**Correctness:** 3
**Technical Novelty And Significance:** 3
**Empirical Novelty And Significance:** 3
**Recommendation:** 3

**Clarity, Quality, Novelty And Reproducibility:**

The paper is well written and the proposed method is easy to follow. Because I’m not familiar with the field of temporal logic point process, I cannot evaluate the novelty and reproducibility of this work.


**Strength And Weaknesses:**

Overall I think the paper presents an interesting idea of applying RL to solve combinatorial optimization problems in temporal logic rule learning. I’m not familiar with the field of temporal logic point process (TLPP) and the related works in this field, so I will focus on the optimization method part in this review.

### Pro

The paper is well presented and the proposed method is easy to follow. The authors include a detailed introduction about the preliminaries and make it easy for readers to understand the problem. Although I’m not familiar with TLPP, I am able to quickly understand its formulation and the optimization problem associated with learning TLPP thanks to the high quality introduction of this paper.


The empirical results of the proposed method seem strong. The proposed method outperforms many baselines in real datasets.


### Con

While the problem of searching for logic rules can certainly be framed as an RL problem, I’m not convinced that this is a good formulation, and I want to argue that formulating it as an RL problem here makes it unnecessarily more difficult. First of all, the problem has no inherent Markov state transition structure of an MDP because all tokens are directly generated by the agent, unlike in an MDP, where the states can only be influenced by the actions of the agent rather than directly chosen by the agent. Moreover, the reward here is the result of an entire sequence rather than the result of a single time step. Combining these two characteristics, I believe the problem is better formulated as a contextual bandit problem, and solved using a bandit algorithm instead of an RL algorithm. Using a bandit algorithm will directly remove the long horizon credit assignment difficulty of RL and likely result in much stabler training and better performance.


Given that the optimization problem is a standard back box optimization problem, it would be important to compare to many widely used back box optimizer baselines. Some examples of these algorithms include cross-entropy methods, genetic algorithms and simulated annealing methods. There are many well-developed software packages for these blackbox optimization algorithms that can be easily integrated into the specific problem, so it would be important to try them out and include them in baseline comparisons.


**Summary Of The Paper:**

This paper focuses on the problem of learning the intensity functions for temporal point processes (TPP) to model the occurrence of events in irregular time intervals. Specifically, the authors consider the representation of the intensity functions as parameterized by an exponentiated weighted sum of kernels switched by temporal logic rules. Given this parameterization, the overall objective is to find a set of rules and their corresponding weights to maximize the likelihood of data.

To optimize this objective, the authors break the MLE problem into two steps, a master step and a substep. In the master step, the weights are optimized via convex optimization. In the substep, the logic rules are selected. In order to optimize in the combinatorial space of logic rules, the authors formulate the problem as an RL problem, where the policy outputs a token of logic rules at a time and receives a reward when the sequence of tokens is completed. The authors then apply the risk seeking policy gradient method to solve this problem.

The paper includes empirical evaluations of the proposed method on synthetic and real datasets, and the results suggest that the proposed method achieves good likelihood and absolute error.


**Summary Of The Review:**

While this paper presents an interesting method of applying RL to optimize the log likelihood of TLPP, I’m not really convinced that RL is the right approach and I believe that simpler approaches could work better for this problem. Hence, I cannot recommend acceptance of this paper in its current state. I highly encourage the authors to try applying bandit methods and including some black box optimization baselines during the author response phase.

---

> ### Author Response · Authors · 2022-11-16
> **In response to reviewer XGxF**
>
> Thank you for raising your concerns about using RL in rule search! We would like to clarify the motivation of RL as follows.
>
> $\star\star\star \textbf{RL v.s. simpler approaches like contextual bandit}$
>
> We claim that the “temporal logic rule” has a sequential nature like language, which is very different from the “ordinary logic rules”, where the body predicates are permutation invariant. For temporal logic rules, the content and structure of temporal logic rules are more like language, and the selection of the previous predicate will affect the selection of the latter predicates.
>
> For example, the selected predicates in the rule will influence the types of temporal relations to be considered in the same rule. There exist dependencies between predicates and their temporal relations.
>
> For another example, in applications like healthcare, a complete temporal logic rule may include predicates about “Symptoms”, “Treatment” and “Clinical Detection Indicators”. The selected predicates about “Symptoms” may affect the latter selection of predicates belonging to the “Treatment” or “Clinical Detection Indicators” predicates. This corresponds to some long-range dependency among predicates with a rule.
>
> It is thus more suitable to use the neural RL policy to generate rule content (rather than the contextual bandits) to capture the intrinsic short-range and long-range temporal dependency of predicates with a rule. The interdependence among predicates in selection can also be memorized by neural RL policy in training, which also enables the policy transfer to speed up the rule search in similar tasks, which is another merit of the neural RL policy.
>
> Also, from the practical view, sequentially generating predicates using RL policy makes it easier (by the adoption of dynamic masks in generation) to satisfy the pre-specified rule templates as domain/prior knowledge constraints.
>
> Overall, we say that it has a clear motivation to use RL policy as opposed to simpler policies like contextual bandits in the temporal logic rule learning setting.
>
> $\star\star\star \textbf{Regrading the reward of RL}$
>
> We’ve made clarification about the reason why we use RL policy. The terminal reward challenge is the price that we need to pay. It is a “common problem” faced by most structure learning problems using RL, like Neural Architecture Search (NAS) problems. In fact, the mainstream methods of NAS exactly use a terminal reward and obtain reasonable performance. Here we provide the link of the a classic NAS paper and we also cite it in our paper:
>
> $\textbf{Reference}$
>
> https://arxiv.org/abs/1611.01578
>
> $\star\star\star \textbf{Regarding the comparison with contextual bandits}$
>
> We have thought carefully about the feasibility of using contextual bandits (a simple method as claimed by the reviewer) in our setting.  We conclude that using contextual bandits to learn rules is also a hard problem. By formulating our problem as a contextual bandit problem, we build some reward surface: $r_f \sim r_{\theta}(\cdot)$, where $r_\theta(\cdot)$ is the function of the combinatorial predicates and their temporal ordering relations. $\theta$ are the parameters we want to learn from the reward. The reward $r_f$ is evaluated by computing the current log-likelihood of the data (combined with a rule complexity penalty term) under the formed rules (i.e., bandit). In this setting, temporal logic rules would be formulated by the combinatorial predicates instead of sequentially generating.
>
> In our paper, our method can discover multiple rules with various lengths to explain data, where the number of rules and their rule lengths can be automatically determined by our algorithm. If we want to achieve the same outcome using contextual bandits, we have to (at least) formulate this problem as a top-$K$ combinatorial bandit problem, where $K$ out of $N$ bandits are chosen at each round, and their holistic reward (i.e., the current log-likelihood function combined with penalty under $K$ rules) are to be computed.  Top-$K$ combinatorial bandit problem is inherently challenging to solve. What’s more, in our setting, $N$ is very large due to the combinatorial search space of the predicates and their temporal relations. Enumerating the potential combinatorial predicates and their temporal relations would be inefficient or even intractable.

---

> > ### Comment · Reviewer_XGxF · 2022-11-18
> > **Re: In response to reviewer XGxF**
> >
> > I'd like to thank the reviewer for the detailed response.
> >
> > ### Comparison to Black-Box Optimization Method
> > First of all, I still believe it is important to at lease include some black-box optimization algorithms as baselines for this paper. For example, it is well known that genetic algorithms can be applied to code generation (genetic programming), and I believe the authors could easily take an exiting implementation and replace just the RL part of the proposed method to try it on this problem. In this way the overall two phase framework is preserved and the RL vs back-box optimization can be fairly compared. Given that these algorithms have been used in similar problems for decades, I think a good paper should include at lease some of these results for comparison.
> >
> > ### RL vs Contextual Bandit
> > I believe the sequential structure of the generated rules is not the same as the Markov temporal structure in MDPs, since all the tokens in the rules sequence can be directly controlled by the policy. In a broader sense of contextual bandit settings, the action space can also be a variable length sequence and in fact the policy can be parameterized by the same RNN as in this paper. The real difference from RL is that contextual bandit algorithms is not optimizing the cumulative rewards but a single step reward, which often makes it much easier to use. Some examples of modern instantiations of these ideas for sequence optimization are CbAS/DbAS [1, 2] and P3BO [3].
> >
> >
> > Overall I still believe that comparing to black-box optimization and contextual bandit methods are necessary, so I will keep my evaluation of the paper.
> >
> >
> >
> > ## References
> > [1] Brookes, David, Hahnbeom Park, and Jennifer Listgarten. "Conditioning by adaptive sampling for robust design." International conference on machine learning. PMLR, 2019.
> >
> > [2] Brookes, David H., and Jennifer Listgarten. "Design by adaptive sampling." arXiv preprint arXiv:1810.03714 (2018).
> >
> > [3] Angermueller, Christof, et al. "Population-based black-box optimization for biological sequence design." International Conference on Machine Learning. PMLR, 2020.

---

> > > ### Author Response · Authors · 2022-11-18
> > > **In response to reviewer XGxF**
> > >
> > > Thank you for the nice references you provided! We agree that the suggested experiments as baselines can further improve the quality of our paper. We will focus on these baselines and design corresponding algorithms to apply the mentioned black-box optimization methods and the contextual bandit method to our temporal point process problem.

---

> ### Author Response · Authors · 2022-11-16
> **In response to reviewer XGxF**
>
> $\star\star\star \textbf{Regarding the comparison of other black-box optimization methods.}$
>
> We have also thought about the feasibility of using other classic black-box optimization algorithms in our setting. We think these methods cannot be directly adapted to the temporal logic rule discovery problem without modification or special designs.
>
> The major reason comes from that the black-box methods need to accommodate the following unique properties brought about by the temporal logic rule set discovery: 1) there exist intrinsic sequential nature in the appearance of predicates and their relations within individual rules. 2) the ground-truth rules may be multiple, various lengths, and of varying importance. In fact, our rule learning algorithm is motivated by these properties. Each rule (content) is sequentially generated by RL to capture the temporal dependency, and the rules are added according to their importance scores in a controlled manner.
>
> Our arguments for each method are as follows:
>
> (1) $\textbf{Cross-entropy methods:}$ the basic idea is, given a parameterized probability distribution $h(x;\theta)$ and an objective function $r_x$, which can be considered as the reward of a temporal logic rule, to iterate between sampling temporal logic rule $f\sim h(x;\theta)$ from this probability distribution and optimizing this distribution. We can start with random parameters $\theta$.  Then sample rules $f\sim h(x; \theta)$ and sorts them by their rewards $r(x)$ in increasing order. At a last step, we adjust the parameters such that better solution $x$ which thus had a larger reward $r(x)$, get a higher probability $h(x;\theta)$. This method seems appealing, but it is hard to sample predicates to form a temporal logic rule that has a sequential nature that are exactly in the correct order to form a temporal logic rule.
>
> (2) $\textbf{Genetic algorithms:}$ we can start from several initial rules, then continue to combine new predicates with genetic variation (i.e., combining rules with modifications), recalculate each rule’s reward, keep rules with higher rewards, and continue to iterate genetic variation until every reward of rules in our rule set no longer increases. The final remaining rules with the highest rewards are the uncovered rules. However, the genetic algorithm is a heuristic method. It cannot theoretically guarantee that the optimal solution can be obtained but can only obtain the optimal solution with a certain probability. Our subproblem is principal and uses the dual price (Eq.7 in our paper) that can be exactly guaranteed that we can find the optimal solution using the complementary slackness theorem. It also guarantees that every optimization we make is no worse than the last. And the unimportant rules can be downweighed automatically. This has been proved by following works which we provided as references.
>
> (3) $\textbf{Simulated annealing methods:}$ we can model the temporal logic rules as binary numbers (e.g., a $K\times n$ matrix with binary elements, where $K$ is the number of rules and $n$ is the total number of predicates and temporal relations). Then keep cooling down based on the initialization temperature (number, namely the rules), and jump out of the current optimal solution with a certain probability. Cooling may produce positive optimization (accept) or negative (accept with a certain probability). Cooling in the temporal logic rule scenario is to change the binary number corresponding to the predicate combinations. However, this method also cannot generate temporal logic rules with sequential nature, and the total number of rules needs to be pre-specified.
>
> Moreover, the above-mentioned methods cannot be easily transferred to new tasks. In contrast, our neural-RL policy can memorize the inter-predicate-group dependency in temporal logic rule construction and this can be transferred to similar tasks.

---

### Official Review · Reviewer_AuUW · 2022-10-29

**Confidence:** 4
**Correctness:** 4
**Technical Novelty And Significance:** 2
**Empirical Novelty And Significance:** 2
**Recommendation:** 6

**Clarity, Quality, Novelty And Reproducibility:**

Clarity: good
Quality: good
Reproducibility: authors give enough materials in the paper itself and in complementary material.  It may be possible to reproduce, but I haven't checked

**Strength And Weaknesses:**

The work is quite interesting and shows good results in the
application of predicting sepsis from the MIMIMC-III database by
distinguishing time predictions for LowUrine from NormalUrine rules. Various state of the
art systems were compared against the method proposed in this work.

I wonder if classical methods based on (Probabilistic) Inductive Logic
Programming wouldn't work as well as your proposed method. Why isn't a
method like that compared in your work? Is it very different? TTE
problems may be also modeled using ILP and variants that are
neurosymbolic.

Why did you use only one application? How well would your method work
if you used other examples?

Why do you concentrate on grounded terms and not on first order logic terms with a logical variable?

**Summary Of The Paper:**

This work presents a reinforcement temporal logic rule learning
algorithm to jointly learn temporal logic rules (before-like kind of
rules) and their weights from event data. The proposed learning
algorithm alternates between a rule generator stage and a rule
evaluator stage, where a neural search policy is learned by
risk-seeking gradient descent to discover new rules in the rule
generator stage.

**Summary Of The Review:**

This work presents a reinforcement temporal logic rule learning
algorithm to jointly learn temporal logic rules (before-like kind of
rules) and their weights from event data. The proposed learning
algorithm alternates between a rule generator stage and a rule
evaluator stage, where a neural search policy is learned by
risk-seeking gradient descent to discover new rules in the rule
generator stage.

The work is quite interesting and shows good results in the
application of predicting sepsis from the MIMIMC-III database by
distinguishing LowUrine from NormalUrine rules. Various state of the
art systems were compared against the method proposed in this work.

I wonder if classical methods based on (Probabilistic) Inductive Logic
Programming wouldn't work as well as your proposed method. Why isn't a
method like that compared in your work? Is it very different? TTE
problems may be also modeled using ILP and variants that are
neurosymbolic.

Why did you use only one application? How well would your method work
if you used other examples?

---

> ### Author Response · Authors · 2022-11-15
> **In response to reviewer AuUW**
>
> We are grateful for your careful reading and constructive suggestions. Your suggestions will help us further improve the quality of the paper.
>
> We will address your concerns one by one as follows.
>
> $\star\star\star \textbf{What’s the difference between our method with the Probabilistic ILP?}$
>
> Thank the reviewer for pointing out the (potential) use of ILP to address our problem. This is an interesting idea. No such work has ever been done for the temporal point process data. We will highlight the technical issues that might arise to illustrate why ILP cannot be easily or directly used “in our setting”.
>
> ILP would require sufficient positive and negative facts regarding various temporal ordering constraints of the events as inputs. ILP model will face a significant scalability challenge to have an accurate performance.  As a contrast, our method uses raw event times as the algorithm inputs, without the need to extract and prepare the temporal relation facts as inputs. The temporal relation will be automatically discovered by the RL type of search algorithm.
>
> $\star\star\star \textbf{Could we design a neuro-symbolic variant in our setting?}$
>
> This is also an interesting angle. Our model is not neuro-symbolic at this stage. Designing a neuro-symbolic variant for our temporal logic rule learning method will be an advanced task, which is equally interesting and deserves lots of future investigation. These are some technical issues that we may need to take care of when we design a neuro-symbolic variant. Neuro-symbolic methods often work on a continuous relaxation of this discrete space. We may need to careful hand-design or introduce task-specific rule templates to narrow down the hypothesis space to achieve a good performance. A neural-symbolic variant may face scalability issues coming from the huge amount of event sequences. But we agree with the reviewer that a neural-symbolic variant can have many merits that our current model cannot have. It is an interesting future work.
>
> $\star\star\star \textbf{Why concentrate on grounded terms and not on first order logic terms with a logical variable?}$
>
> Our rule-learning framework is based on the following probabilistic sequential event modelling framework “Temporal Logic Point Processes”. Li. et al. ICML 2020
> We all made the same assumption:  the states of the predicates can be fully observed, and the randomness comes from the occurrence times or state transition times of the predicates.
> This is aligned with the general temporal point process modelling ideas, where the inter-event time intervals are modeled as random variables, the event states are completely observed, and the intensity function can capture the long-range historical dependency. Allowing event states being latent will make the problem more challenging -- we want to leave it as our future work.

---

> ### Author Response · Authors · 2022-11-16
> **In response to reviewer AuUW**
>
> $\star\star\star \textbf{Another application beyond healthcare}$
>
> Following your suggestion, we have added another interesting application in our experiment.
>
> $\textbf{Application Description:}$
>
> We aim to understand shoppers’ purchase patterns given their eye fixation event data.
> Our $\textbf{conjecture}$ is: the location of the items, shopper-assessed values of the items, and the shopper's visual habits (usually looking from left to right) will affect their final item choice. We learned temporal logic rules and their weights to quantitatively understand this.
>
> $\textbf{Dataset Reference:}$
>
> https://github.com/fredcallaway/optimal-fixations-simple-choice/tree/master/data/krajbich_PNAS_2011
>
> $\textbf{Dataset Description:}$
>
> Three items randomly placed on the “left”, “middle” and “right” on the supermarket shelf, each has a unique “price” (value).
> Each shopper evaluated three items by eye fixation until they identified an item to purchase.
> The data record each shopper’s eye fixated items, when and where, and their final purchased item. There are 30 participants, each with at most 100 independent trials. At each trial, participants were asked to look at these three items and choose the item that they think is most valuable. There are 2966 trials in total. On average, for one trial, a participant has 4.3011 eye fixations.
>
> $\textbf{Predicate Definition:}$
>
> We are interested in explaining three final choices of a shopper:
>
> 1) finally choose the item with the actual (not shopper-assessed) largest value
>
> 2) finally choose the item with the last eye fixation, and
>
> 3) finally choose the item with the longest eye fixation.
>
> These final choices define the head predicate set.
> Another 18 predicates are about the location of the items, value of the items, and the time of eye fixation of a shopper on one specific item. Please refer the table below for a complete predicate set.
>
> ```
> --------------------------------------------------------------------------
>  Eye Fixation                                | Left_MaxValue_LongFixation
>  (template: “location_value_fixation time”)  | Left_MaxValue_ShortFixation
>                                              | Left_MidValue_LongFixation
>                                              | Left_MidValue_ShortFixation
>                                              | Left_MinValue_LongFixation
>                                              | Left_MinValue_ShortFixation
>                                              | Middle_MaxValue_LongFixation
>                                              | Middle_MaxValue_ShortFixation
>                                              | Middle_MidValue_LongFixation
>                                              | Middle_MidValue_ShortFixation
>                                              | Middle_MinValue_LongFixation
>                                              | Middle_MinValue_ShortFixation
>                                              | Right_MaxValue_LongFixation
>                                              | Right_MaxValue_ShortFixation
>                                              | Right_MidValue_LongFixation
>                                              | Right_MidValue_ShortFixation
>                                              | Right_MinValue_LongFixation
>                                              | Right_MinValue_ShortFixation
> --------------------------------------------------------------------------
> Final Choice                                 |  FinalChoice_LargestValue
>                                              |  FinalChoice_LastFixation
>                                              |  FinalChoice_LongestFixation
> --------------------------------------------------------------------------
> Temporal Relation                            | Before
> --------------------------------------------------------------------------
> ```

---

> > ### Author Response · Authors · 2022-11-16
> > **Discovered rules and result discussion for this new application**
> >
> > $\textbf{Discovered Rules:}$
> >
> > Note: In this eye fixation application, only the temporal relation “Before” needs to be considered. In the table below, we omit “Before”, which is supposed to be there, to simplify the presentation.
> >
> > ```
> > -----------------------------------------------------------------------------
> > Weight | Rule
> > -----------------------------------------------------------------------------
> > 0.0550 | Rule-1: FinalChoice_LargestValue <-- Middle_MaxValue_LongFixation ^
> >        |         Left_MidValue_ShortFixation ^ Right_MinValue_ShortFixation
> > -----------------------------------------------------------------------------
> > 0.0976 | Rule-2: FinalChoice_LargestValue <-- Left_MaxValue_LongFixation ^
> >        |         Middle_MinValue_ShortFixation ^ Right_MidValue_ShortFixation
> > -----------------------------------------------------------------------------
> > 0.0278 | Rule-3: FinalChoice_LastFixation <-- Left_MaxValue_LongFixation ^
> >        |         Middle_MinValue_ShortFixation ^ Left_MaxValue_ShortFixation
> > -----------------------------------------------------------------------------
> > 0.0479 | Rule-4: FinalChoice_LongestFixation <-- Left_MaxValue_LongFixation ^
> >        |         Middle_MidValue_LongFixation ^ Left_MaxValue_ShortFixation
> > -----------------------------------------------------------------------------
> > 0.0648 | Rule-5: FinalChoice_LargestValue <-- Left_MaxValue_LongFixation ^
> >        |         Middle_MidValue_ShortFixation ^ Right_MinValue_LongFixation ^
> >        |         Left_MaxValue_ShortFixation
> > -----------------------------------------------------------------------------
> > 0.0015 | Rule-6: FinalChoice_LastFixation <-- Middle_MidValue_ShortFixation ^
> >        |         Left_MaxValue_LongFixation ^ Middle_MidValue_ShortFixation ^
> >        |         Left_MaxValue_ShortFixation
> > -----------------------------------------------------------------------------
> > 0.0457 | Rule-7: FinalChoice_LongestFixation <-- Left_MinValue_LongFixation ^
> >        |         Middle_MaxValue_LongFixation ^ Right_MidValue_ShortFixation ^
> >        |         Middle_MaxValue_ShortFixation
> > -----------------------------------------------------------------------------
> > 0.0241 | Rule-8: FinalChoice_LargestValue <-- Middle_MidValue_ShortFixation ^
> >        |         Left_MaxValue_LongFixation ^ Right_MinValue_ShortFixation ^
> >        |         Middle_MidValue_LongFixation ^ Left_MaxValue_ShortFixation
> > -----------------------------------------------------------------------------
> > 0.0243 | Rule-9: FinalChoice_LastFixation <-- Middle_MidValue_ShortFixation ^
> >        |         Right_MinValue_ShortFixation ^ Left_MaxValue_LongFixation ^
> >        |         Middle_MidValue_LongFixation ^ Left_MaxValue_ShortFixation
> > -----------------------------------------------------------------------------
> > 0.0145 | Rule-10: FinalChoice_LongestFixation <-- Left_MaxValue_LongFixation ^
> >        |         Middle_MidValue_ShortFixation ^ Left_MaxValue_LongFixation ^
> >        |         Right_MinValue_LongFixation ^ Left_MaxValue_ShortFixation
> > -----------------------------------------------------------------------------
> > ```
> >
> > $\textbf{Result Discussion:}$
> >
> > The above discovered temporal logic rules summarize the eye fixation patterns before shoppers making choices. From the results, we have the following discoveries:
> >
> > 1) the final fixation is shorter
> > 2) the later (but not the final) fixations are longer
> > 3) people are more likely to begin to look from the left or from the middle.
> >
> > Specifically, if a shopper finally chooses the item with the largest value, he may first glance over all three items at least once, or after looking at all three items, go back to check the item he wants to choose, and then make a choice (Rule 1, 2, 5, and 8).
> > And people are more used to looking from left to right (Rule 2, 3, 5, and 7).
> > If a shopper finally chooses the item with the last eye fixation, he may only take a quick look at these items and may miss one or two of these items (Rule 3, 6, and 9).
> > If a shopper finally chooses the item with the longest eye fixation, he may spend a lot of time on most of these items, reevaluating the value of these items back and forth in his mind (Rule 4, 7, and 10).
> > In summary, our discovered temporal logic rules provide insight into shopper’s perchance behaviors in terms of eye fixation patterns.

---

### Decision · Program_Chairs · 2023-01-20

**Decision:**

Reject

**Justification For Why Not Higher Score:**

This was a tough one.  Two of the reviewers were at 6 and two at 3 and there wasn't any movement after engagement with the authors. This is not my area and I had a tough time adjudicating the differences so perhaps erred on the negative. If someone who knows more about TPPs would champion it I could revise upwards.

**Justification For Why Not Lower Score:**

N/A

**Metareview: Summary, Strengths And Weaknesses:**

This paper focuses on the problem of learning temporal logic rules for temporal point processes (TPP). The overall objective is to find a set of rules and their corresponding weights to maximize the likelihood of data.

The MLE problem is solved in two steps, a master step, in which the weights are optimized via convex optimization, and a substep, in which the logic rules are selected. The authors formulate the rule induction problem as an RL problem, where the policy outputs one rule token at each step and receives a reward when the sequence of tokens is completed. The authors then apply a risk-seeking policy gradient method to solve this problem.

Strengths: All the reviewers thought this was an interesting approach and the results seem promising. The writing is reasonably clear and there is sufficient introductory material to help readers, such as myself, who are not expert in TPP.  The paper includes empirical evaluations of the proposed method on synthetic and real datasets, such as medical diagnosis, and the results suggest that the proposed method achieves good likelihood and absolute error.I appreciate the author's engagement during the review process and their running of additional experiments.

Weaknesses: There were questions about competing approaches such as ILP (or neural ILP) techniques that might make reasonable baselines but were not discussed in the paper. The authors argue that they would not be performant but it is perhaps not entirely clear still. There were also doubts raised about the suitability of RL approaches for this problem, given that the underlying formulation relies on history and therefore is non-Markovian.  The authors responded to these concerns but the reviewers were not convinced that other approaches such as bandits might not be more appropriate. Finally, there was a question about how doctors would assess the learned rules (and those that were rejected). The authors mention that doctors found the results useful and agreed with the model's assessment but the reporting of this is anecdotal and perhaps a user study would be appropriate to make the case stronger.

Given the concerns and the current scores I can't recommend acceptance at this point. Clearly this is good work in a difficult setting. I hope the authors take the reviewer concerns into account in a rewrite and resubmit.

**Summary Of Ac-Reviewer Meeting:**

N/A